# Synergistic Integration of HDAC Inhibitors and Individualized Neoantigen Therapy (INT): A Next-Generation Combinatorial Approach for Cancer Immunotherapy

**DOI:** 10.3390/vaccines13060550

**Published:** 2025-05-22

**Authors:** Rui Han, Huiling Zhou, Baoqing Peng, Shasha Yu, Jiajie Zhu, Jiaojiao Chen

**Affiliations:** 1Department of Chinese Medicine Oncology, The First Affiliated Hospital of Naval Medical University, Shanghai 200433, Chinaalexpeng520@163.com (B.P.);; 2Department of Chinese Medicine, Naval Medical University, Shanghai 200433, China; 3Department of Gastroenterology, Tongde Hospital of Zhejiang Province, Hangzhou 310012, China

**Keywords:** histone deacetylase inhibitors, individualized neoantigen therapy (INT), immunotherapy, antigen presentation, tumor microenvironment

## Abstract

Background: Cancer immunotherapy has advanced, yet therapeutic resistance and low response rates remain problematic. This study explores histone deacetylase inhibitors (HDACis) as adjuvants for cancer vaccines to enhance anti-tumor immunity and overcome these challenges. Methods: A comprehensive review of relevant literature was conducted. Studies on the immunomodulatory mechanisms of HDACis, their effects on Individualized neoantigen therapy (INT), and clinical applications were analyzed. Results: HDACis enhance anti-tumor immunity through multiple mechanisms. They activate endogenous retroelements, expanding the “antigen repository”. HDACis also upregulate MHC class I and II molecules, enhance the antigen processing machinery, improve MHC—I complex stability, and remodel the tumor immune microenvironment. Early clinical trials of HDACis combined with peptide vaccines show promising safety and immunological responses. However, challenges exist, such as HDACi-mediated PD-L1 regulation, optimal sequencing strategies, and biomarker development. Conclusions: The combination of HDACis and cancer vaccines has significant potential in cancer immunotherapy. Despite challenges, it offers a new approach to overcome tumor heterogeneity and immune evasion, especially for patients with limited treatment options. Further research on toxicity management, triple-drug combinations, biomarker identification, and delivery systems is needed to fully realize its clinical benefits.

## 1. Introduction

Therapeutic cancer vaccines represent a promising immunotherapeutic approach that stimulates the immune system to recognize and eliminate cancer cells. Individualized neoantigen therapy (INT) as a special type of cancer vaccine has demonstrated certain advantages in clinical practice [1]. The recent success of the KEYNOTE-942 trial, which demonstrated significant improvement in recurrence-free survival for melanoma patients receiving an mRNA-based INT combined with pembrolizumab, has renewed interest in this modality. This trial showed a 44% reduction in risk of recurrence or death compared to pembrolizumab monotherapy, with a recurrence-free survival rate of 78.6% versus 62.2% at 18 months median follow-up [2].

Despite their potential, INT faces several challenges that limits its efficacy as monotherapy. These include tumor heterogeneity, immune evasion mechanisms, immunosuppressive microenvironments, and insufficient presentation of tumor-associated antigens [3]. The tumor microenvironment can suppress vaccine-induced immune responses through regulatory T cells, myeloid-derived suppressor cells, and inhibitory immune checkpoints [4]. Additionally, downregulation of antigen presentation machinery in tumor cells further compromises vaccine effectiveness. These challenges highlight the critical need for adjuvant strategies to enhance INT potency [4].

Several adjuvant approaches have emerged to address these limitations. Toll-like receptor (TLR) agonists like imiquimod and CpG oligodeoxynucleotides enhance innate immune activation but can cause inflammatory side effects [5]. STING agonists demonstrate potent type I interferon induction and tumor regression in preclinical models, though they may trigger autoimmune-like syndromes at higher doses [6]. Cytokines such as GM-CSF and IL-12 boost T-cell responses but have narrow therapeutic windows [7]. When compared with these approaches, histone deacetylase inhibitors (HDACis) potentially offer unique advantages through their ability to simultaneously modify the tumor epigenetic landscape and immune microenvironment with established safety profiles from their approved indications [8].

Histone deacetylases (HDACs) are enzymes that remove acetyl groups from lysine residues on histones and are classified into four major families: Class I (HDAC1,2,3,8), Class II (HDAC4-7, 9-10), Class III (sirtuins, SIRT1-7), and Class IV (HDAC11) [9,10]. HDAC inhibitors (HDACis) increase chromatin accessibility by inhibiting histone deacetylation, thereby altering gene expression patterns [9,10]. Although the FDA has approved several HDACis for hematological malignancies, such as vorinostat for cutaneous T-cell lymphoma, clinical trials of monotherapy in solid tumors have generally yielded unfavorable results [11,12]. Current research primarily focuses on combination strategies, including HDACi with immune checkpoint inhibitors, targeted therapies, or cellular therapies like CAR-T [13].

Our investigations into HDACi’s immunomodulatory effects have revealed their capacity to enhance tumor antigen presentation, promote effector T cell infiltration, and inhibit myeloid-derived suppressor cell function [8]. Despite limited efficacy as monotherapy, HDACis show particular promise as combination partners with INT due to their ability to overcome tumor heterogeneity and immune evasion mechanisms [14,15,16]. The complementary actions of these two modalities—with cancer vaccines providing specific tumor antigens and HDACis enhancing their presentation and recognition—create a compelling rationale for combination therapy.

This review provides the first comprehensive analysis of the mechanistic rationale for integrating HDACi with INT as a next-generation combinatorial approach for cancer immunotherapy. We examine the synergistic effects, early clinical evidence, potential challenges, and future directions of this promising strategy that could transform treatment paradigms for patients with limited therapeutic options.

## 2. Potential Coordination Mechanisms

### 2.1. Synergizing with DCs

Dendritic cells (DCs), as professional antigen-presenting cells, play a central role in cancer immunotherapy, particularly in the process of INT activating the immune system [17]. HDACi exerts complex and multifaceted regulatory effects on DCs with significant dose and time dependency, demonstrating powerful synergistic potential.

Studies indicate that HDACi significantly influences multiple biological functions of DCs and promotes DC maturation. HDACi inhibits HDAC activity, increases acetylation levels of histones H3 and H4, and loosens chromatin structure, thereby facilitating the binding of DC maturation-related transcription factors such as NF-κB, AP-1, and STAT1 to target gene promoter regions, upregulating the expression of CD80, CD86, CD40, and MHC molecules [18]. Notably, different types of HDACi affect DC maturation differently; SAHA (Vorinostat) and TSA (Trichostatin A) primarily promote DC maturation through inhibition of HDAC1 and HDAC2, while MS-275 (Entinostat) tends to selectively inhibit HDAC1 [19,20,21]. Regarding antigen uptake and processing, recent research reveals that HDACi can bidirectionally regulate DCs’ antigen uptake capacity, closely related to HDACi’s temporal window of action [20]. HDACi intervention enhances cytoskeletal dynamics and regulates endocytic receptors (such as Mannose receptors and Scavenger receptors), promoting tumor antigen uptake by DCs; however, prolonged treatment may reduce uptake efficiency [18]. In the antigen processing stage, HDACi enhances antigen cleavage and transport efficiency by upregulating immunoproteasome subunits (such as LMP2, LMP7, and MECL-1) and TAP1/2 transport proteins [22] (Table 1). Particularly, HDACi promotes the activity of lysosomal proteins like Cathepsin S and Cathepsin B, optimizing the MHC class II molecule antigen processing pathway [23]. During antigen presentation, HDACi not only upregulates MHC class II molecules by increasing CIITA (MHC class II transcriptional activator) expression but also expands DCs’ capacity to present lipid antigens by enhancing expression of antigen presentation-associated auxiliary molecules like CD1d, activating broader T cell subsets, including NKT cells [23]. Additionally, HDACi significantly alters DCs’ cytokine secretion profile, promoting production of pro-inflammatory factors like IL-12p70, IL-15, and IFN-α, while inhibiting the release of immunosuppressive factors such as IL-10 and TGF-β, creating a microenvironment more conducive to T cell activation [24]. HDACi also enhances DCs’ ability to migrate to lymph nodes by regulating chemokine receptors like CCR7 and CXCR4, crucial for effectively initiating anti-tumor immune responses [25]. New discoveries in trans-epigenetics reveal that HDACi can indirectly influence DC function by regulating microRNAs (such as miR-155 and miR-146a) expression [26].

In clinical translation research, sequential treatment strategies combining HDACi with DC vaccines (pretreatment of tumor microenvironment with HDACi followed by DC vaccine inoculation) have demonstrated significant synergistic anti-tumor effects in melanoma, pancreatic cancer, and non-small-cell lung cancer models, not only enhancing CD8+T cell infiltration and activity but also reducing the proportion of regulatory T cells (Tregs) and myeloid-derived suppressor cells (MDSCs) in the tumor microenvironment [27]. Notably, in a Phase I/II clinical trial reported in 2023, the combination of Panobinostat with autologous DC vaccines for recurrent glioblastoma patients achieved preliminary positive results, with 30% of patients achieving disease stability or partial remission, and treatment group patients showing marked expansion of tumor-specific CD8+ T cell populations [28]. Nevertheless, HDACi regulation of DC function still faces challenges including narrow dosage windows, insufficient selectivity, and clinical medication sequencing. Future efforts should focus on developing more selective HDACi subtype-specific inhibitors and optimizing combination strategies with immunotherapies to maximize their potential in cancer immunotherapy.

### 2.2. Activation of Endogenous Retroelements: Expanding the “Antigen Repository”

Endogenous retroelements are specialized DNA sequences within biological genomes that originated from retroviral infections. These elements integrated their genetic material into host genomes and persisted throughout evolutionary history [29]. Histone deacetylase inhibitors (HDACis) can modulate the chromatin structure of endogenous retrovirus (ERV) regions in tumor cells by regulating histone deacetylation levels, transforming their relatively condensed state into a more relaxed configuration, thereby influencing ERV transcriptional activity [30].

HDACis have been discovered to activate multiple cryptic transcription start sites of LTR12 elements (a type of ERV) in various cancer cells, particularly the LTR12C subtype [31] (Table 1). For instance, in the PC3 prostate cancer cell line, LTR12C functions as a promoter for the long non-coding RNA SchLAP1, whose overexpression affects the anti-tumor activity of the SWI/SNF chromatin modification complex. In p53-mutated liver cancer, LTR12C-derived lncRNA PRLH1 can also interfere with tumor cell proliferation. These newly expressed antigens can be recognized by the immune system, increasing the variety and quantity of tumor antigens. Additionally, ERV activation may influence tumor cell immunogenicity, making them more susceptible to immune system recognition and attack [32,33,34]. In infection-related research, LTR12-mediated transcription can also be triggered in uninfected bystander cells, potentially protecting cells by inducing an antiviral state. This suggests that LTR12 activation may initiate a series of immune response-related changes, which could similarly apply to tumor cells, making them more easily recognized by the immune system [35,36,37] (Figure 1).

Evidence has shown that in colorectal cancer models, dual inhibition therapy targeting DNA methyltransferase and histone deacetylase (C02S, targeting HDAC1, 2, 3, 8) not only upregulates tumor ERV expression but also activates viral mimicry responses through the MDA5–MAVS signaling pathway. This remodels the tumor immune microenvironment, enhances immune cell infiltration, and significantly improves the anti-tumor effects of anti-PD-L1 monoclonal antibodies [38]. Furthermore, in triple-negative breast cancer cells, dual inhibition of HDAC and DNMT has been found to increase ERV expression, producing double-stranded RNA (dsRNA). The intracellular pattern recognition receptor MDA5 recognizes dsRNA, activating the MAVS pathway and subsequently TBK1. TBK1 phosphorylates IRF7, promoting the formation of active transcription complexes that initiate type I and III interferon expression, triggering viral mimicry responses and further activating innate immunity. The resulting type I interferons activate NK cells and CD8+ T cells, enhancing NK cell cytotoxicity against tumor cells and promoting CD8+ T cell activation. This enables specific recognition and lysis of triple-negative breast cancer cells, directly reducing tumor cell numbers. Type III interferons regulate immune cell functions within the tumor microenvironment. Simultaneously, the release of cytokines and chemokines in the tumor microenvironment attracts more immune cells to the tumor site, inhibiting tumor cell proliferation, invasion, and metastatic capabilities, creating an environment unfavorable for tumor cell growth [39,40] (Figure 1).

Additionally, HDACi can directly influence HERVs (Human Endogenous Retroviruses) sites in T cells, thereby modulating anti-tumor immunity (Table 1). For example, the White research team found that vorinostat (an HDACi targeting Class I and II HDACs: HDAC1, 2, 3, 6, 8, 10) intervention can regulate over 2000 individual HERV sites on CD4+ T cells [41]. Another study demonstrated that SAHA intervention can significantly modulate 722 endogenous retroviral sites on T cells, with the 5 most upregulated ERV sites belonging to the ERV3, HERV-W, MER4, and ERVL families [42]. These loci contain open reading frames or can encode proteins, potentially becoming tumor-associated antigens. The immune system can recognize these abnormally expressed antigens, activating immune cells such as T cells and enhancing their cytotoxic effects against tumor cells. Moreover, upregulated ERVs can be recognized by Pattern Recognition Receptors (PRRs), activating innate immune responses. HERV-W RNA may activate RIG-I-like receptors (RLRs) or Toll-like receptors (TLRs), initiating downstream signaling pathways that induce the production of interferons (IFNs) and other cytokines, enhancing both antiviral and antitumor immune responses, as well as strengthening immune surveillance [43] (Figure 1).

Therefore, combining HDACi with INT could potentially enhance viral mimicry and innate immune activation through upregulated ERVs, potentially increasing tumor cell sensitivity to INT and promoting tumor cell apoptosis [43] (Figure 1). However, the specific synergistic effects depend on multiple factors and require further exploration and research.

### 2.3. Enhancement of Antigen Processing Machinery

HDACi not only upregulates the expression of MHC molecules themselves but also enhances various related antigen processing components [44].

Evidence has reported that Trichostatin A (TSA, a HDACi targeting Class I and Class II HDACs: HDAC1-10) can significantly upregulate the expression of TAP2 (Transporter Associated with Antigen Processing 2), LMP2 (Low Molecular Weight Protein 2), LMP7 (Low Molecular Weight Protein 7), and Tapasin in melanoma cell lines (B16F10 cells) [45]. Additionally, MHC class I genes and TAP1 (Transporter Associated with Antigen Processing 1) gene, which were initially expressed at low levels, showed substantial increases after TSA treatment. Similarly, intervention with another HDAC inhibitor (Valeric acid, VA) also significantly increased the expression levels of LMP, TAP, Tapasin, and class I genes in this cell line. Furthermore, the expression of class I molecules in B16F10 cells, as well as CD40 and CD86, were simultaneously elevated, helping to enhance the cells’ ability to process and present antigens through the MHC class I pathway, thereby activating CD8^+^ T cells and strengthening immune responses [45,46].

LMP2 and LMP7 are essential components of the immunoproteasome [47]. Compared to constitutive proteasomes, immunoproteasomes are more efficient at generating antigenic peptides suitable for binding to MHC class I molecules. When LMP gene expression levels increase, more immunoproteasomes form, and LMP can also alter the cleavage specificity of proteasomes, favoring the production of antigenic peptides with high affinity for MHC class I molecules [48]. These antigenic peptides, produced intracellularly, provide abundant substrates for subsequent binding to MHC class I molecules and antigen presentation, thereby enhancing the function of the antigen processing machinery [49].

The heterodimer formed by TAP1 and TAP2 is responsible for transporting antigen peptides generated in the cytoplasm to the endoplasmic reticulum (ER), where these peptides bind to newly synthesized MHC class I molecules [50] (Table 1). When TAP gene expression is upregulated, its capacity to transport antigenic peptides increases, allowing more peptides to be transported to the ER and form complexes with MHC class I molecules, thereby enhancing antigen presentation efficiency [50]. Efficient TAP transport ensures sufficient antigenic peptides in the ER to bind with MHC class I molecules, maintaining continuous antigen presentation [50]. Insufficient TAP expression prevents effective entry of antigenic peptides into the ER, resulting in inadequate binding of peptides to MHC class I molecules, thus affecting antigen presentation and immune activation [50].

Tapasin plays a crucial “molecular chaperone” role in the ER, connecting MHC class I molecules to TAP and promoting proper folding and stable binding of antigenic peptides with MHC class I molecules [51]. When Tapasin gene expression levels increase, more Tapasin proteins function to optimize the binding process between MHC class I molecules and antigenic peptides, improving the efficiency and quality of this binding, and enhancing the ability of antigen processing components to present antigens on the cell surface [51]. Tapasin contributes to the formation of stable peptide–MHC class I molecule complexes, which, after transport to the cell surface, exist more stably and are recognized more efficiently by T cells [51]. These stable complexes can continuously activate T cells and enhance immune responses, further highlighting Tapasin’s key role in enhancing antigen processing component functions.

Regarding class I genes encoding MHC class I molecules, increased expression levels mean more MHC class I molecules on the cell surface [52]. A greater number of MHC class I molecules can bind more antigenic peptides and present them on the cell surface, increasing opportunities for recognition by CD8+ T cells, thereby enhancing the immune system’s surveillance and killing capacity against tumor cells or infected cells [52]. Evidence suggests that overexpression of class I genes in tumor cells can enhance tumor cell sensitivity to CD8+ T cells, promoting immune cell killing of tumor cells. The increase in MHC class I molecule-antigenic peptide complexes allows the immune system to more precisely identify abnormal cells, enhancing the effectiveness of immune defense and thus achieving better tumor suppression [52].

INT exerts its anti-tumor effects by presenting antigens, while HDACi enhances the function of antigen processing components, undoubtedly providing more theoretical support for their combined use [53]. After INTs enter the body, they translate tumor antigens; HDACi enhances antigen processing component functions, forming more immunoproteasomes [54]. These immunoproteasomes not only more efficiently generate antigenic peptides suitable for binding to MHC class I molecules but also alter cleavage specificity to produce high-affinity binding peptides, allowing tumor antigens produced by INT to be processed more efficiently, providing abundant substrates for antigen presentation [54]. Simultaneously, HDACi can upregulate relevant genes to enhance the ability of TAP1 and TAP2 to transport antigenic peptides to the ER for binding with MHC class I molecules, enabling more efficient transport and binding of antigenic peptides generated from tumor antigens expressed by INT, improving antigen presentation efficiency, and increasing opportunities for CD8^+^ T cell recognition [44,54]. Furthermore, HDACi can optimize the binding process between antigenic peptides and MHC class I molecules and strengthen their stability, ensuring sustained T cell activation [55].

### 2.4. Upregulation of MHC Class I Molecule Expression

Evidence has revealed that SAHA (an HDACi targeting Class I HDACs: HDAC1, 2, 3, 8; Class II HDACs: HDAC4-9) can rapidly induce STAT1 and Smad2/3 phosphorylation in NSCLC cells, leading to increased expression of p-STAT1 and p-Smad2/3 in the nucleus (promoting their nuclear translocation). This enhances the binding of p-Smad2 to the promoters of HLA-A, HLA-B, and HLA-C, thereby increasing MHC expression while simultaneously elevating the percentage of CD8^+^ T cells, the frequency of IFN-γ^+^ CD8^+^ T cells, and IFN-γ levels [56]. SAHA has also been found to significantly enhance the expression of MHC I-related genes (HLA-A, HLA-B, HLA-C), peptide transport genes (TAP1 and TAP2), and co-stimulatory molecules CD80 and CD86 in A549, H520, and Lewis lung cancer cells [56].

In glioma cells (U251, GL261), SAHA upregulates cell surface MHC-I expression and increases MHC-I density. SAHA also significantly upregulates the expression of four antigen presentation-related proteins—TAP1, TAP2, LMP2, and LMP7—in U251 and GL261 cells, enhancing immune cell recognition of tumor cells [57]. Furthermore, HDAC inhibitors can promote MHC class I molecule expression by increasing H3K27ac and H3K9ac levels in the MHC class I gene promoter regions, facilitating the recruitment of transcription factors such as STAT1 and IRF1, and enhancing RNA polymerase II binding [58]. They can also restore the dynamic balance of acetylation/deacetylation co-regulated by HDACs (histone deacetylases) and HATs (histone acetyltransferases), thereby restoring the antigen presentation function of MHC and enhancing immune surveillance and clearance of tumor cells [59].

OBP-801, a selective inhibitor targeting class I HDACs (Yakult HDAC inhibitor, HDAC1-3), has been found to upregulate the expression of the immunoproteasome subunit LMP2 in clear cell renal cell carcinoma (ccRCC) cell lines RENCA, 786-O, and Caki-1 in a dose-dependent manner. It also increases the expression of MHC class I molecules on the cell surface. Gene knockdown experiments have shown that downregulating the LMP2 gene inhibits OBP-801’s ability to upregulate MHC class I molecule expression. Additionally, OBP-801 intervention is accompanied by increased MHC class I molecule expression and a rise in the proportion of CD45^+^CD3e^+^ T cells among tumor-infiltrating lymphocytes. Research has found that MHC class I molecule presentation positively correlates with the proportion of CD45^+^CD3e^+^ T cells and negatively correlates with tumor growth rate. Moreover, evidence indicates that OBP-801 can be combined with anti-PD-1 antibodies and enhance antitumor effects by increasing MHC class I molecule expression [60] (Figure 2).

The fundamental action of HDAC inhibitors is to suppress histone deacetylase activity, which normally removes acetyl groups from histone tails [61]. When HDACs are inhibited, the activity of histone acetyltransferases (HATs) is relatively enhanced. The acetylation levels at sites such as H3K9, H3K14, H3K27, H4K5, H4K8, and H4K16 increase. The chromatin structure in the promoter regions of MHC class I genes becomes more open [62,63,64]. Additionally, proteins containing bromodomains (BRDs) recognize acetylated histones. These BRD proteins act as “scaffold proteins”, recruiting transcriptional co-activators, forming transcription initiation complexes, including the recruitment of RNA polymerase II [65].

HDACi significantly enhances tumor cell antigen presentation capability by upregulating MHC class I molecules and related antigen presentation genes [44,66]. MHC molecules play a crucial role in tumor vaccine therapy, especially in the context of INT. INT expresses and presents tumor-specific antigens (TSAs) or tumor-associated antigens (TAAs) to the immune system by encoding them, but their efficacy highly depends on the ability of MHC class I molecules to present antigenic peptides to CD8^+^ T cells [67]. HDACi ensures efficient presentation of antigens encoded by mRNA vaccines by enhancing the expression and function of MHC class I molecules, thereby activating stronger T cell immune responses [45,46,49]. Furthermore, HDACi reverses tumor cell immune evasion and restores immune system surveillance functions by promoting nuclear translocation of transcription factors such as STAT1 and Smad2/3 and increasing histone acetylation levels, further activating MHC class I molecule expression, providing a broader antigen presentation foundation for tumor vaccines [68,69,70]. Simultaneously, HDACi can increase CD8^+^ T cell infiltration and function (such as IFN-γ secretion) in the tumor microenvironment and synergize with immune checkpoint inhibitors (such as anti-PD-1 antibodies) to enhance antitumor effects (Figure 2) [44,71].

Based on these mechanisms, combination therapy with HDACi and mRNA INT not only enhances vaccine immunogenicity and overcomes tumor heterogeneity and immunosuppressive microenvironments but also induces long-lasting tumor-specific immune memory, significantly enhancing the intensity and durability of antitumor immune responses. This provides a rational and effective synergistic strategy for tumor treatment. Therefore, research in this field requires more attention and investment.

### 2.5. Improvement of MHC-I Complex Stability

Evidence has found that HDACi can enhance peptide transport into the endoplasmic reticulum by increasing histone acetylation at TAP gene promoters, activating STAT1 and IRF1 transcription factors, enhancing interferon signaling pathways, and improving TAP protein stability and assembly efficiency, thereby upregulating TAP1/TAP2 transporter expression and function [45,72,73]. Additionally, they improve MHC-I complex assembly and folding by upregulating molecular chaperone proteins (such as calnexin, tapasin, ERp57), optimizing the ER environment, promoting high-affinity peptide generation, and regulating protein ubiquitination [45,71,74,75,76,77]. Simultaneously, HDACi can also modulate extracellular vesicle composition by regulating vesicular protein components, altering MHC-I content in exosomes, modifying vesicle surface proteins, and regulating proteins involved in cellular secretory pathways, thereby changing the quantity, content, and function of extracellular vesicles, ultimately enhancing MHC-I complex stability on the cell surface and intercellular transport [78] (Table 1).

β2M (β2-Microglobulin) is a crucial component of MHC class I molecules, playing a vital role in their proper assembly, stable expression on cell surfaces, and normal functional performance [79]. Recent studies have shown that Vorinostat and Butyrate (targeting HDAC1, 2, 3, 8) can significantly increase the expression levels of β2M and HLA-class I in various Colorectal Cancer-Cancer Initiating Cells (CRC-CICs). This has been demonstrated in multiple cell populations, including CRC-CICs isolated from patient biopsy tissues, primary CICs provided by other researchers (CIC-2, 3, 5, 7), and established cell lines from ATCC including SW620 and SW480. This enhanced expression strengthens the antigen-presenting capability of tumor cells, enabling better recognition by T cells and increasing IFN-γ release, ultimately leading to a more robust anti-tumor immune response [80,81,82]. Reduced β2M/MHC class I expression has been observed in various tumor cells, including melanoma and malignant Hodgkin Reed-Sternberg (HRS) cells. The ability of HDAC inhibitors to regulate β2M and consequently improve MHC class I molecule expression shows considerable potential in initiating anti-tumor immunity [83,84,85,86].

### 2.6. Enhancement of MHC Class II-Mediated Antitumor Immunity

MHC class II complexes consist of heterodimeric glycoproteins with α and β transmembrane chains that together create an antigen-binding cleft between their α1 and β1 domains. The principal role of these molecules involves capturing peptide antigens and displaying them to CD4+ T lymphocytes, which triggers subsequent immune cascades and cellular responses [87]. HLA-DM, a non-classical MHC class II molecule, plays a crucial role in facilitating peptide loading onto MHC-II’s antigen-binding groove, essential for antigen presentation. HDAC inhibitors function by suppressing histone deacetylases, modifying chromatin structure, and consequently regulating gene expression, particularly in MHC-II-related immune modulation and cancer immunotherapy [88].

In the transcriptional regulation of MHC-II expression, CIITA (II Major Histocompatibility Complex Transactivator) serves as a key transcription factor. It recruits transcription factors and other co-regulatory proteins, alters chromatin interactions, and modifies the local epigenetic landscape to promote MHC-II gene expression. HDAC inhibitors (HDACis) activate the MHC-II pathway’s antitumor immune response by inhibiting histone deacetylases, thus maintaining lysine acetylation on histones [71]. For instance, research has shown that IFN-γ induces transcription at the CIITA-pIV site in colon cancer cells, while treatment with TSA induces transcription at the CIITA-pIII site. Combined treatment with both further enhances CIITA induction and MHC-II expression [89,90]. Additionally, HLA-DM plays a role in loading antigenic peptides onto MHC-II, ensuring that MHC-II effectively presents antigens and activates immune cells [89,90]. Furthermore, Sodium Valproate (VPA), an HDACi, has been found to reverse the suppression of MHC II expression mediated by DcR3 (Decoy Receptor 3) and restore antitumor immune responses [91]. DcR3, a member of the tumor necrosis factor receptor superfamily, is overexpressed in malignant tumors such as pancreatic cancer. Its overexpression has been linked to ERK/cJNK-related histone deacetylation, which leads to decreased expression of MHC II-associated genes in monocyte-derived macrophages (MDMs), reducing the expression of MHC II complexes and thus inhibiting the antitumor immune response (Figure 3) [92].

INT can encode tumor-associated antigens, which, after being taken up by antigen-presenting cells, express the tumor antigen. These antigens bind to MHC class II molecules to form a complex, which is then presented on the cell surface to activate CD4+ T cells, triggering an antitumor immune response [93]. HDACi upregulates MHC class II molecule expression by inhibiting histone deacetylases. When combined, the enhanced expression of MHC class II molecules induced by HDACi improves the efficiency of tumor antigen presentation, activating more CD4+ T cells. These activated cells secrete cytokines that help activate CD8+ T cells and other immune cells, thus forming a robust antitumor immune network. Moreover, HDACi can reverse the immune evasion mechanism by which tumor cells downregulate MHC class II molecule expression. When used together with INT, HDACi enhances the recognition and attack of tumor cells. Additionally, the flexibility of mRNA technology and the adjustable dosage of HDACi allow for better adaptation to individual patient differences, enabling personalized treatment (Figure 3). Research has already confirmed the potential of HDACi in modulating immune molecule expression and the antitumor capacity of INT, providing clinical support for the feasibility of their combined use in cancer therapy [94].

### 2.7. Other Impacts on the Immune Microenvironment

Chemokine CXCL8 (C-X-C motif chemokine ligand 8), also known as Interleukin-8 (IL-8), is a pro-inflammatory chemokine that facilitates the recruitment of neutrophils to areas of inflammation, infection, or injury. However, in the context of cancer, CXCL8 is produced by various cell types within the tumor microenvironment (TME), including infiltrating immune cells, stromal cells, and tumor cells [95,96]. Recent studies have revealed several new crosstalk mechanisms between CXCL8 and components of the tumor microenvironment (TME), which can suppress anti-tumor immunity, promote tumor progression, and even create positive feedback loops [95,97]. In various cancers, CXCL8 is typically overexpressed and associated with poor prognosis [97,98]. For instance, there is evidence suggesting that higher levels of CXCL8 correlate with increased infiltration of myeloid-derived suppressor cells (MDSCs) in the immune microenvironment, leading to reduced T-cell activity [99]. A study involving patients with melanoma, non-small cell lung cancer (NSCLC), and renal cell carcinoma (RCC) indicated that elevated baseline CXCL8 levels in plasma were linked to poorer clinical outcomes in patients receiving immunotherapy (such as nivolumab or ipilimumab). This suggests that CXCL8 could serve as a biomarker to predict the effectiveness of ICI therapy [100] (Figure 4).

CXCR2 (C-X-C chemokine receptor type 2) is one of the main receptors for CXCL8 and belongs to the CXC chemokine receptor family. It is also commonly overexpressed in patients with various cancers and is recognized as a marker of poor prognosis in several malignancies [101,102,103]. For example, a study by Jack Leslie’s team utilized an HCC mouse model and induced HCC in the context of non-alcoholic steatohepatitis (NASH) through a high-fat, high-sugar diet, simulating the complex immune environment of chronic liver damage and tumors. The study used the CXCR2 small molecule inhibitor AZD5069 to directly intervene with the downstream targets of the CXCL8/CXCR2 axis, both as a monotherapy and in combination with anti-PD-1. When AZD5069 was used alone, the migration of neutrophils in the liver was inhibited. However, combining AZD5069 with anti-PD-1 significantly reduced tumor burden and extended the survival time of the mice. This combination therapy notably increased the activation of XCR1+ dendritic cells and the number of CD8+ T cells in the tumor, both of which are associated with anti-tumor immunity. Further experiments showed that when myeloid cell recruitment was inhibited by genetic methods, or the XCL1 ligand was neutralized, or CD8+ T cells were depleted, the therapeutic effect was significantly reduced. Moreover, under combination treatment, tumor-associated neutrophils (TANs) switched from a pro-tumor to an anti-tumor phenotype. These reprogrammed TANs formed immune cell clusters in direct contact with CD8+ T cells, which were enriched with anti-tumor granzyme B, indicating a synergistic interaction between TANs and CD8+ T cells, enhancing the anti-tumor effect. Therefore, inhibiting CXCR2 reduces the pro-tumor effect of neutrophils and enhances the effectiveness of anti-PD-1 immunotherapy by weakening the immune suppression in the tumor microenvironment, boosting T cell activity, and effectively controlling tumor progression [104]. Additionally, Han ZJ’s team investigated the impact of the CXCL8–CXCR2 axis on the tumor microenvironment (TME) and immune therapy through multiple in vivo and in vitro experimental models. The experiments were divided into a control group, a CXCL8/CXCR1/2 inhibitor treatment group, and a CXCR1/2 inhibitor combined with anti-PD-1 immune checkpoint inhibitor group. The results showed that CXCL8, through binding to CXCR1/2, activates the PI3K/AKT and MAPK signaling pathways, enhancing the migration and invasion capabilities of tumor cells. In mouse models, blocking CXCR1/2 significantly reduced the recruitment of neutrophils and myeloid-derived suppressor cells (MDSCs) to the tumor microenvironment, thereby reducing immune suppression and enhancing the anti-tumor immune response. Particularly when combined with anti-PD-1 therapy, CXCR1/2 antagonists significantly reduced tumor burden and prolonged the survival time of the mice. Immune microenvironment analysis revealed that blocking the CXCL8/CXCR1/2 pathway reduced the infiltration of MDSCs and tumor-associated neutrophils while enhancing the infiltration and activity of CD8+ T cells. The study concluded that CXCL8 promotes tumor cell migration, invasion, and immune evasion through the CXCR1/2 axis, and blocking this signaling pathway can significantly improve the efficacy of immune checkpoint inhibitors, especially when used in combination with anti-PD-1, showing a pronounced anti-tumor effect [98]. Therefore, inhibitors targeting the CXCL8-CXCR1/2 axis hold promise as an important strategy to enhance the effectiveness of cancer immunotherapy (Figure 4).

The activation of the CXCL8/CXCR2 signaling pathway is believed to suppress anti-tumor immunity, promote tumor cell invasiveness, and facilitate immune evasion. As a result, this pathway not only serves as a potential biomarker for prognosis in cancer patients but also represents a highly promising target for immunotherapy [101,102,103]. More importantly, recent evidence suggests that the expression of the CXCL8/CXCR2 axis can be regulated by HDAC3 [105,106]. Team of Lili has demonstrated through chromatin immunoprecipitation (ChIP) analysis that HDAC3 occupies the promoter region of the CXCL8 gene and inhibits its transcriptional activity [105]. Evidence also found that in tumor cells with HDAC3 knockout, the expression of CXCL8 was significantly upregulated. Additionally, the researchers used HDAC inhibitors (such as RGF966 and Entinostat) to treat the cells and measured the secretion levels of CXCL8 protein via ELISA. They observed that HDAC3 inhibition significantly reduced CXCL8 secretion. Finally, in an in vivo mouse model, the researchers assessed CXCL8 expression and immune cell infiltration in the tumor microenvironment using immunohistochemistry and flow cytometry. This included increased numbers of CD4+ and CD8+ T cells, a higher proportion of effector T cells (such as CD8+ T cells secreting IFN-γ) in the tumor, and an increase in antigen-presenting cells (such as CD11c+ dendritic cells). These findings further demonstrated that HDAC3 inhibitors could enhance the anti-tumor immune response by reducing CXCL8 expression [105,106] (Figure 4).

In studies on mouse colon adenocarcinoma cells MC38 and fibrosarcoma tumor cells MCA205, Li and colleagues found that HDAC3 inhibitors (such as vorinostat and romidepsin) can suppress the expression of CXCL9, CXCL10, and CXCL11 by directly affecting the binding in the promoter region. This, in turn, recruits T cells to infiltrate the tumor microenvironment (TME) and inhibits tumor growth [105]. There is also evidence suggesting that the inherent inhibition of HDAC8 in HCC cells can enhance enhancer reprogramming, epigenetically activating the production of CCL4 (C-C motif ligand 4). The absence of CCL4 either in vitro or in vivo in NOD/SCID mice does not affect cell proliferation but leads to a significant reduction in CD8+ T cell migration. However, downregulation of HDAC8 expression can promote CD8+ T cell infiltration within the tumor. Studies have shown that the selective HDAC8 inhibitor PCI-34051, when used alone, exhibits inhibitory effects on liver cancer. When combined with anti-PD-L1 antibody treatment, it significantly improves tumor eradication and induces the generation of memory CD8+ T cells, which produce a lasting immune response and extend survival in tumor-bearing mouse models. These findings suggest that HDAC8 inhibitors have potential clinical applications in liver cancer treatment and, when combined with immune checkpoint blockers, can enhance the post-treatment anti-tumor immune response [107] (Figure 4).

Therefore, HDAC inhibitors can improve anti-tumor immunity through multiple mechanisms, thereby synergistically enhancing the anti-cancer effects of mRNA INT. This combination strategy of epigenetic regulation and immune modulation has the potential to reshape the tumor immune microenvironment, potentially creating favorable conditions for mRNA INT.

## 3. Downregulation of HDACs Using Genome Editing Mechanisms

The evolution of genome editing technologies has revolutionized our approach to HDAC downregulation, offering unprecedented precision and versatility. DNA-targeting platforms such as CRISPR-Cas9, Prime Editing, and TALENs, alongside RNA-targeting systems like CRISPR-Cas13, each present distinct advantages tailored to specific research objectives [108]. Moreover, deactivated variants (dCas9 and dCas13) enable reversible regulation without permanent genomic alterations, proving invaluable for investigating HDAC functions and developing therapeutic strategies [109]. Moving forward, research priorities should include enhancing specificity and efficiency, minimizing off-target effects, and creating more sophisticated delivery systems to facilitate clinical translation. Integrating bioinformatics with high-throughput screening methodologies to identify optimal targets will enable precise HDAC expression control, potentially transforming treatment approaches for epigenetic disorders and cancer.

### 3.1. CRISPR-Cas9 Mediated Gene Editing

CRISPR-Cas9 (Clustered Regularly Interspaced Short Palindromic Repeats and CRISPR-associated protein 9) represents a revolutionary breakthrough in genome editing technology. The mechanism begins with transcription of the CRISPR region to generate crRNA, which then combines with tracrRNA to form single guide RNA (sgRNA) [110]. This complex guides the Cas9 endonuclease to specific genomic sequences, creating double-strand breaks (DSBs) in the DNA. Cellular repair mechanisms address these breaks through either homology-directed repair (HDR) or non-homologous end joining (NHEJ) [111].

To effectively downregulate HDAC genes, researchers can design sgRNAs targeting coding regions or regulatory elements, directing Cas9 to cleave the DNA at these locations. When NHEJ repair occurs, it frequently introduces frameshift mutations, deletions, or insertions that compromise gene function or substantially reduce expression [112]. Evidence have demonstrated that successfully disrupting specific HDAC gene regions using CRISPR-Cas9 not only diminishes expression of the targeted HDAC but also influences associated signaling networks and cellular functions [113].

Interestingly, HDACi can enhance the efficiency of advanced CRISPR-Cas9 systems. Research indicates that inhibiting HDAC1 and HDAC2 significantly improves both knockout and knock-in efficiency. This enhancement occurs because HDACi treatment relaxes chromatin structure, creating more opportunities for Cas9 to access target DNA. Specifically, compounds like Entinostat and Panobinostat increase H3 histone acetylation in chromatin, amplifying CRISPR-Cas9 editing capabilities [113].

### 3.2. Transcriptional Repression Mediated by Deactivated Cas9 (dCas9)

Deactivated Cas9 (dCas9), engineered to lack nuclease activity while retaining DNA-binding capacity, offers an elegant approach to HDAC regulation. When guided to HDAC gene promoters or transcription initiation sites, dCas9 can dramatically reduce expression by physically blocking RNA polymerase, ribosome binding, or transcription factor interactions [114]. This technique, termed CRISPR interference (CRISPRi), avoids DNA breaks, thereby circumventing potential off-target complications and chromosomal rearrangements associated with traditional CRISPR-Cas9 applications [115].

Researchers have successfully employed CRISPRi to target critical regulatory elements within HDAC promoter regions, effectively suppressing specific HDAC subtypes [116]. Furthermore, the fusion of transcriptional repression domains (particularly KRAB) to dCas9 substantially enhances inhibitory effects, yielding more pronounced reduction in HDAC expression levels. This approach shows remarkable promise for therapeutic applications [117].

### 3.3. Prime Editing Technology

Prime Editing emerges as a sophisticated genome editing platform offering novel strategies for HDAC downregulation. This technology ingeniously combines modified Cas9 with reverse transcriptase, guided by specialized pegRNA (prime editing guide RNA), enabling precise nucleotide substitutions, insertions, or deletions without generating double-strand breaks [118]. For manipulating HDAC genes, Prime Editing can introduce specific mutations in functional domains or regulatory elements—such as premature stop codons, splice site alterations, or promoter mutations—affecting transcription, translation, or splicing to achieve targeted downregulation [119]. Evidence suggests that introducing specific mutations in HDAC gene promoter regions can disrupt transcription factor binding sites, inhibiting gene expression and ultimately reducing protein levels [120]. The exceptional precision and reduced off-target effects position Prime Editing as particularly promising for clinical applications.

### 3.4. TALEN Technology

TALEN (Transcription Activator-Like Effector Nuclease) technology comprises three key components: an N-terminal domain containing nuclear localization signals, a central domain with TALE repeat sequences recognizing specific DNA motifs, and a C-terminal domain featuring FokI nuclease activity [121]. By designing TALENs specifically targeting HDAC genes, researchers can direct FokI to create double-strand breaks at precise locations [121].

When cellular repair mechanisms address these breaks, mutations or deletions often result, effectively downregulating HDAC expression. Compared to CRISPR-Cas9, TALENs offer enhanced specificity and fewer off-target effects, despite their more complex design requirements [122]. Studies have documented successful TALEN-mediated downregulation of various HDAC subtypes, with corresponding epigenetic modifications and biological consequences.

### 3.5. RNA Interference Technology and CRISPR-Cas13 System

RNA interference (RNAi) directly targets HDAC transcripts, inhibiting protein expression by specifically binding and degrading mRNA through short interfering RNA (siRNA) or short hairpin RNA (shRNA) [115]. Evidence has confirmed that siRNAs or shRNAs targeting specific HDAC subtypes effectively reduce corresponding expression levels and alter critical cellular processes including proliferation, differentiation, and apoptosis [123].

The CRISPR-Cas13 system represents an innovative RNA-targeting approach for HDAC downregulation. Unlike DNA-targeting CRISPR-Cas9, Cas13 specifically targets RNA molecules [124]. Researchers can design guide RNAs targeting HDAC transcripts, enabling Cas13 to bind and degrade these mRNAs, thereby reducing HDAC protein expression [124]. Recent advances have introduced deactivated Cas13 (dCas13) for translational-level CRISPR interference (Tl-CRISPRi) [125]. This methodology does not cause mRNA degradation but instead prevents protein synthesis by blocking translation. Optimized guide RNAs direct dCas13 to HDAC mRNA translation initiation regions, preventing ribosome binding or elongation, thereby precisely controlling HDAC protein levels. This technology has successfully achieved simultaneous downregulation of multiple targets, demonstrating exceptional specificity and versatility [126].

### 3.6. Switchable Cas12a-Based System

An innovative switchable system based on Cas12a presents a novel tool for HDAC activity monitoring and regulation. By incorporating specific lysine acetylation modifications to Cas12a protein, researchers can initially suppress its nuclease activity; when HDACs remove these acetylation marks, Cas12a function is restored [127]. This characteristic has been cleverly adapted to develop sensitive HDAC activity assays while providing new approaches for targeted HDAC downregulation [128]. Theoretically, conditional Cas12a systems could be engineered to activate in specific cellular environments, downregulating target HDACs only under predetermined conditions.

## 4. Clinical Applications of HDACi Combined with Tumor Vaccines

### 4.1. Polypeptide Vaccine PVX-410 Combined with Citarinostat

Current clinical research on the combination of HDACi and tumor vaccines for cancer treatment remains limited, with most studies in early-phase clinical trials. For example, a Phase Ib clinical trial (NCT02886065) targeting smoldering multiple myeloma (SMM) is currently underway (Table 1). SMM is an asymptomatic clonal plasma cell proliferative disorder, intermediate between monoclonal gammopathy of undetermined significance (MGUS) and multiple myeloma (MM) [129].The study protocol includes two arms: the first arm combines a peptide-based vaccine (PVX-410) with Citarinostat (CC-96241), a small-molecule oral histone deacetylase inhibitor; the second arm employs a triple-drug regimen of PVX-410, Citarinostat, and lenalidomide (Table 2).

Separately, a Phase 1/2a clinical trial (NCT01718899) reported that the PVX-410 vaccine enhanced anti-tumor immunity in SMM patients by increasing the proportion of tetramer-positive and interferon-γ-producing CD3+CD8+ T cells [130]. Another in vitro study demonstrated that Citarinostat reduced the production of Th2 cytokines (e.g., IL-4, IL-5, IL-6, IL-10, IL-13) in melanoma models, enhanced cytotoxic activity in mixed lymphocyte reactions, and downregulated the Tregs/FOXP3 axis, thereby potentiating T-cell-mediated anti-tumor responses in melanoma [131].

Currently, the study results of NCT02886065 have not been submitted. This could be due to the study not being finished, the submission deadline not being reached, the study not needing result submission, or the sponsor/investigator receiving approval to delay submission.

### 4.2. Multi-Drug Combination Regimens Beyond Dual Therapy

a. Combinationof H1299 Cell Lysate Vaccine, Entinostat (HDACi), Nivolumab, and Montanide(R) ISA-51 VG Adjuvant

A Phase I/II clinical trial (NCT05898828) aimed to evaluate the safety of a quadruple regimen comprising the H1299 cell lysate vaccine (Table 2), Entinostat (a histone deacetylase inhibitor, HDACi), Nivolumab (an anti-PD-1 antibody), and Montanide(R) ISA-51 VG adjuvant (an immunostimulatory adjuvant) in patients with advanced esophageal cancer (EsC). However, this study was withdrawn due to insufficient patient enrollment (Table 2).

Previous clinical research (NCT02054104) demonstrated that the H1299 lysate vaccine reduced the percentage of regulatory T cells (Tregs) and downregulated PD-1 expression on Tregs (*p* = 0.0027) in patients with primary thoracic malignancies, thereby enhancing anti-tumor immunity [132]. Another study revealed that Entinostat amplifies antigen-stimulated T-cell responses by suppressing immunosuppressive cell populations, including Tregs and monocytic myeloid-derived suppressor cells (M-MDSCs), further potentiating anti-tumor immunity [133].

b. Combination of BN-Brachyury Vaccine, M7824, T-DM1, and Entinostat

A multi-drug regimen combining BN-Brachyury vaccine, M7824 (a novel bifunctional fusion protein), T-DM1 (an antibody-drug conjugate), and Entinostat (an HDAC inhibitor) has been investigated for metastatic breast cancer in a clinical trial (NCT04296942) (Table 2). The BN-Brachyury vaccine is a recombinant poxviral vaccine targeting the transcription factor brachyury, which is overexpressed in advanced cancers and associated with drug resistance, epithelial–mesenchymal transition (EMT), and metastatic potential. This trial was terminated due to emerging safety concerns related to M7824 and slow patient enrollment (Table 2). Since only one participant was included in this study, the overall response rate could not be calculated. The progression/recurrent time of this included participant was approximately 5 months and 17 days.

A previous Phase I study (NCT04134312) reported that BN-Brachyury vaccine administration in advanced solid tumor patients induced CD4+ and CD8+ T-cell responses in 69% of patients. Notably, 88% and 64% of patients exhibited CD4+ and/or CD8+ T-cell reactivity against cascade antigens CEA and MUC1, respectively, which were not encoded by the vaccine. These T-cell responses were dose-dependent, suggesting the vaccine’s potential to activate anti-tumor immunity even in advanced disease [134].

In vitro studies demonstrated that Entinostat reduces phosphorylation of STAT3 and NFκB, thereby downregulating immunosuppressive downstream targets such as IL-6, IL-10, and Nox2. This mechanism enhances anti-tumor immune efficacy by alleviating immunosuppression [135].

## 5. Current Issues and Challenges

### 5.1. Impact on PD-L1 Expression

While immunotherapy targeting programmed death-1 (PD-1) and its ligand PD-L1 has yielded remarkable clinical outcomes across various tumor types [136], only a subset of patients achieve durable responses. PD-L1 nuclear translocation has emerged as a critical factor limiting therapeutic efficacy [137]. Beyond its conventional immunosuppressive role at the plasma membrane, PD-L1 enhances tumor cell anti-apoptotic capabilities, promotes mTOR activity, and regulates glycolytic metabolism [138].

Research demonstrates that PD-L1 translocates to cell nuclei where it modulates inflammatory and immune responses, promotes tumor invasiveness and metastasis, and triggers expression of immune checkpoint molecules unaffected by PD-1/PD-L1 blockade, resulting in acquired resistance to immunotherapy [137,139]. Gao’s team conducted comprehensive studies using CD274 knock-off tumor cells for chromatin immunoprecipitation and sequencing (ChIP-seq), revealing that nuclear PD-L1 specifically triggers gene expression in immune response pathways. Nuclear PD-L1 positively correlates with immune response-related transcription factors including STAT3, RelA (p65), and c-Jun40, interacting with these factors on DNA to influence anti-tumor immunity [137].

Further molecular analyses revealed that reduced PD-L1 expression leads to downregulation of genes associated with immune surveillance evasion, such as PDCD1LG2 (encoding PD-L2), VSIR (encoding VISTA), and CD276 (encoding B7-H3). This confirms that nuclear PD-L1 upregulates multiple immune checkpoint genes in tumor cells, contributing to resistance against PD-L1/PD-1 blockade therapy [137]. Evidently, PD-L1 in the cell nucleus enhances the activation of multiple immune response pathways, thereby evading immune surveillance [137].

The nuclear translocation of PD-L1 is regulated by acetylation at the Lys 263 site in its C-tail, controlled by HDAC2 inhibitors [137]. Huntingtin-interacting protein 1-related protein (HIP1R) specifically interacts with the PD-L1 C-tail to initiate nuclear translocation. Both HIP1R expression and its binding capacity to PD-L1 depend on Lys 263 acetylation levels, which are regulated by HDAC2 expression [137,140,141].

Upon nuclear entry, PD-L1 interacts with DNA to regulate transcription of antigen presentation-related genes (including MHC-I-related genes) and inflammation pathway-related genes (including IFN-I-related genes). It simultaneously increases expression of other immune checkpoint genes like PD-L2 and VISTA, enhancing cytotoxic T lymphocyte exhaustion and impairing PD-L1 blockade efficacy [137]. More importantly, HDAC2 inhibitors not only inhibit nuclear PD-L1 translocation, enhancing the efficacy of PD-1/PD-L1 checkpoint inhibitors, but also significantly increase the proportion of CD8+ and CD8+GranB (granzyme B)+ cells within tumor-infiltrating lymphocytes. These inhibitors improve the CD8+ cytotoxic T cell to regulatory T cell (CD4+FOXP3+) ratio, modulate cytokine levels (including IL-4, IFN-γ, and TNF-α), and enhance the tumor immune microenvironment [142].

In uveal melanoma (UM) cells, HDAC2 inhibitors restore PD-L1 acetylation, prevent its nuclear entry, and inhibit p-STAT3 binding to the EGR1 promoter region, reducing EGR1 expression and suppressing angiogenic capacity. This suggests that combining anti-PD-L1 immunotherapy with HDAC2 inhibitors could attenuate tumor angiogenesis in UM, offering a novel therapeutic approach [143]. Studies in melanoma reveal that low expression of MIIP and high expression of its downstream molecule HDAC6 correlate with poorer overall survival, with a positive correlation between HDAC6 and PD-L1 protein expression. Experimental evidence indicates that the HDAC6/STAT3 axis regulates PD-L1 expression, suggesting that targeting the MIIP/HDAC6/PD-L1 pathway might enhance immune checkpoint inhibitor efficacy in melanoma [144,145].

HDAC inhibitors that restore PD-L1 expression on tumor cells may potentially undermine mRNA cancer vaccine effectiveness [146]. Prolonged exposure to elevated PD-L1 levels might cause functional exhaustion of vaccine-induced T cells, diminishing their killing capacity [147]. Moreover, prolonged exposure to high levels of PD-L1 might lead to functional exhaustion of vaccine-induced T cells, reducing their killing capacity [148]. Additionally, PD-L1 may interfere with antigen-presenting cell function, affecting the effective presentation of mRNA vaccine-delivered antigens.

Despite these challenges, promising optimization strategies exist. Tumors with elevated PD-L1 expression may respond particularly well to combination therapy involving both mRNA cancer vaccines and PD-1/PD-L1 inhibitors [149]. Such combinations could overcome immunosuppression within the tumor microenvironment, enhancing therapeutic efficacy [149]. HDAC inhibitors could serve as “pretreatment” to ensure high PD-L1 expression on tumor cells, followed by sequential or concurrent administration of mRNA vaccines alongside PD-1/PD-L1 inhibitors. Through deeper mechanistic understanding, more effective triple combination immunotherapeutic approaches may improve clinical outcomes for cancer patients. Moreover, immune responses triggered by mRNA vaccines potentially increase IFN-γ levels within the tumor microenvironment, subsequently inducing elevated PD-L1 expression [150,151]. This feedback mechanism suggests that dynamic changes in PD-L1 expression could serve as a clinically relevant predictive biomarker.

### 5.2. Optimization of Safety and Toxicity Management Strategies

When HDACis are used in combination with mRNA vaccines, excessive stimulation of the anti-tumor immune response may lead to adverse reactions. Cytokine release syndrome (CRS), a severe systemic inflammatory response caused by overactivation of the immune system [152], poses a risk with this combination strategy. CRS is characterized by the accumulation of large quantities of cytokines in the bloodstream (e.g., IL-6 and IFN-γ), accompanied by symptoms such as fever, hypotension, and respiratory distress; in severe cases, it can be life-threatening [153]. Therefore, the safety and toxicity associated with this combined approach require in-depth and systematic evaluation, with the development of targeted intervention systems. For example, to better prevent and manage CRS, comprehensive baseline assessments should be conducted before treatment, including detailed medical history collection, physical examinations, blood routine tests, liver and kidney function analyses, and immune function evaluations, to understand the patient’s baseline health status and identify potential risk factors. During treatment, vital signs should be closely monitored, and interventions such as nutritional support, respiratory and circulatory support, administration of corticosteroids, acetaminophen, antihistamines, and treatments like hemofiltration or continuous renal replacement therapy should be initiated promptly upon detecting abnormalities. After treatment, follow-up monitoring of laboratory and imaging indices should be performed to further optimize toxicity management strategies.

### 5.3. Investigation of Optimal Sequencing in Combination Therapy

Exploring the order of drug administration aims to maximize the anti-tumor efficacy of the regimen. Prior studies have demonstrated that the timing of combined anti-cancer therapy significantly impacts treatment outcomes [154,155]. Rational sequencing of HDACi and mRNA vaccines can ensure tumor cells are exposed to the drugs at their most sensitive stages. In sequential administration, the leading agent may alter the tumor microenvironment (e.g., vascular permeability, hypoxia, immune cell infiltration), enhancing the penetration and efficacy of the subsequent drug [156]. Additionally, the leading agent may transiently inhibit drug resistance mechanisms or activate specific signaling pathways, creating a therapeutic window for the follow-up agent. Advantages of sequential administration include leveraging the sensitivity window induced by the first drug, reducing overlapping peak toxicities, and allowing adjustment of subsequent therapy based on early responses [157]. Limitations include longer treatment cycles and potential reduced efficacy due to primary drug resistance [158]. In concurrent administration, multiple targets can be inhibited simultaneously, minimizing compensatory pathway activation and enhancing immediate synergistic effects, while simplified dosing may improve patient compliance. However, concurrent use may increase treatment-related toxicity [159]. Thus, the specific benefits of sequencing in this regimen require further investigation and evaluation.

### 5.4. Incorporating a Third Agent

Multi-drug combination strategies are a cornerstone of anti-tumor therapy, with frequent clinical use of combinations involving immunotherapies, adjuvants, targeted agents, and chemotherapy [160]. As summarized above, the combination of HDACi and mRNA vaccines shows strong clinical potential [2]. Therefore, a triple strategy integrating HDACi, tumor vaccines, and ICIs may exert stronger anti-tumor immunity through multi-target, multi-pathway mechanisms. Additionally, exploring novel anti-cancer modalities such as Carbonic anhydrase XII inhibitors (CAXIIis), CAR-T, or CAR-M cells as potential third agents to form new treatment strategies warrants urgent investigation to determine if they can further enhance therapeutic efficacy

### 5.5. Development of Biomarkers for Patient Stratification

Identifying relevant biomarkers is critical for precisely selecting patients who will benefit from the HDACi-mRNA vaccine combination. Studies have reported that sarcomas and gastrointestinal stromal tumors with high HR23b protein expression exhibit greater sensitivity to the HDACi vorinostat [161]. Additionally, a study identified five immune subtypes (IS1–IS5) in pancreatic cancer patients, among which those with IS4 and IS5 subtypes exhibited stronger anti-tumor immune responses following vaccination with mRNA vaccines [162]. Therefore, investigating and validating biomarkers that predict a response to this combination will significantly enhance clinical benefits for candidate patients.

### 5.6. Mechanistic Studies of Long-Term Immune Memory Formation

In-depth exploration of the mechanisms by which the HDACi-mRNA vaccine combination promotes durable immune memory is essential for achieving long-term tumor control and reducing recurrence. Future research could involve comprehensive analyses of T cell subsets at different time points post-treatment to track the differentiation trajectory of naïve T cells into memory T cells, identifying key genes and signaling pathways involved in this process [163]. Additionally, investigating the impact of combination therapy on epigenetic modifications in memory T cells will clarify how it regulates gene expression to influence memory T cell formation and maintenance [164]

### 5.7. Development of Novel Delivery Systems

Current delivery systems for HDACi and mRNA vaccines include liposomes, nanoparticles, polymer micelles, exosomes, and viral vectors. Liposomes offer good biocompatibility and can encapsulate hydrophobic HDACis, with surface modification improving targeting [165]; however, they suffer from poor stability (prone to oxidation/hydrolysis), storage challenges, and potential immune-related side effects from certain lipid components [165]. Nanoparticles (e.g., gold, silica) have high drug-loading capacity, but inorganic materials (e.g., metal NPs) may exhibit long-term toxicity, and organic NPs are complex to synthesize and scale up [166]. Polymer micelles can encapsulate HDACi in their hydrophobic cores, have long half-lives, and enable co-delivery of synergistic agents, but their drug-loading capacity is low, and degradation products like polylactic acid may be toxic [167]. Exosomes, with natural membrane structures avoiding immune clearance and inherent tumor-homing ability (including blood–brain barrier penetration), face challenges in isolation, low drug-loading efficiency, and large-scale production [168,169]. Viral vectors offer high transfection efficiency and natural tropism but have limited payload capacity (only small mRNAs) and carry risks of insertional mutagenesis [170]. Given the potential immune risks and side effects of current delivery systems, optimizing a single carrier to co-encapsulate both drugs is highly promising. Recent studies have demonstrated that hydrogel/nano-adjuvant systems can co-deliver whole tumor cells and dendritic cell-based vaccines, significantly increasing effector T cell infiltration, relieving intratumoral immunosuppressive microenvironments, and maximizing immune efficacy [171]. Thus, researching shared carriers for this regimen holds great potential and requires urgent exploration, alongside efforts to reduce costs through technical optimization.

## 6. Conclusions

The synergistic mechanisms uncovered in this work offer a strong theoretical foundation for this novel combinatorial approach. HDAC inhibitors substantially enhance INT efficacy through multi-level mechanisms: they activate endogenous retroelements to expand the “antigen repository”, upregulate MHC class I and II expression, strengthen the antigen processing machinery, improve MHC-I complex stability, and favorably remodel the tumor immune microenvironment via modulation of critical chemokine pathways.

Early clinical investigations of HDAC inhibitors with peptide vaccines have shown promising safety profiles and immunological responses. While challenges persist, including HDACi-mediated PD-L1 regulation and determining optimal treatment sequences, they present opportunities for further investigation rather than insurmountable barriers. Research priorities include developing targeted toxicity management protocols, investigating triple-drug combinations, identifying predictive biomarkers, and engineering innovative delivery systems. The HDAC inhibitor–INT combination offers distinct advantages in addressing fundamental limitations of current cancer immunotherapies. By simultaneously enhancing antigen presentation and creating an immunostimulatory tumor microenvironment, this approach may overcome tumor heterogeneity, immune evasion, and limited response rates. This strategy holds particular promise for patients with “cold” tumors traditionally resistant to immunotherapy, potentially expanding the beneficiary population. Repurposing existing FDA-approved HDAC inhibitors as vaccine adjuvants provides a pragmatic path to clinical implementation with reduced development timelines. This innovative integration of epigenetic modulation with targeted immunotherapy represents a paradigm-shifting strategy that could significantly improve outcomes for cancer patients with limited therapeutic options.

## Figures and Tables

**Figure 1 vaccines-13-00550-f001:**
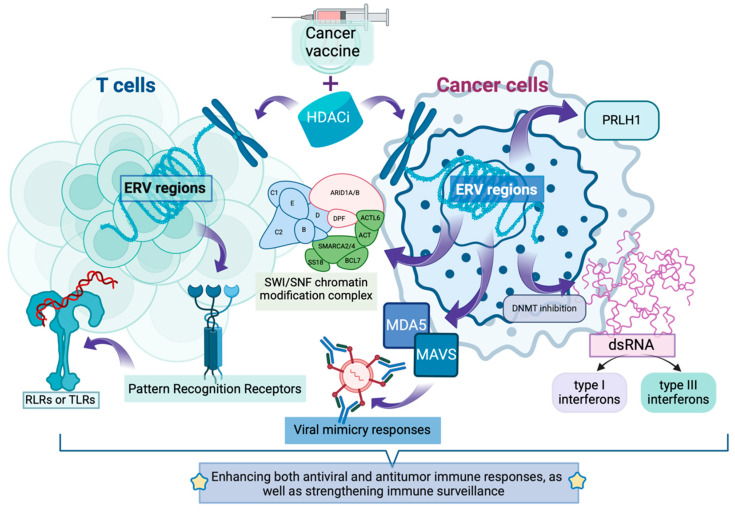
**INT and HDACi synergistically activate endogenous retroelements to expand the antitumor immunogenic repertoire.** HDACi can help INT to induce activation of endogenous retroelement (ERV) regions in both T cells and cancer cells. In cancer cells, HDACi relaxes chromatin structure at ERV loci, particularly at LTR12 elements, inducing expression of transcripts such as PRLH1. DNMT inhibition further enhances ERV expression, generating double-stranded RNA (dsRNA). These dsRNAs are recognized by pattern recognition receptors including MDA5, which activates MAVS signaling to trigger viral mimicry responses. This pathway leads to production of type I and type III interferons. Concurrently, in T cells, HDACi modulates ERV expression, enhancing recognition by RIG-I-like receptors (RLRs) or Toll-like receptors (TLRs). The SWI/SNF chromatin modification complex mediates these regulatory effects on ERV expression.

**Figure 2 vaccines-13-00550-f002:**
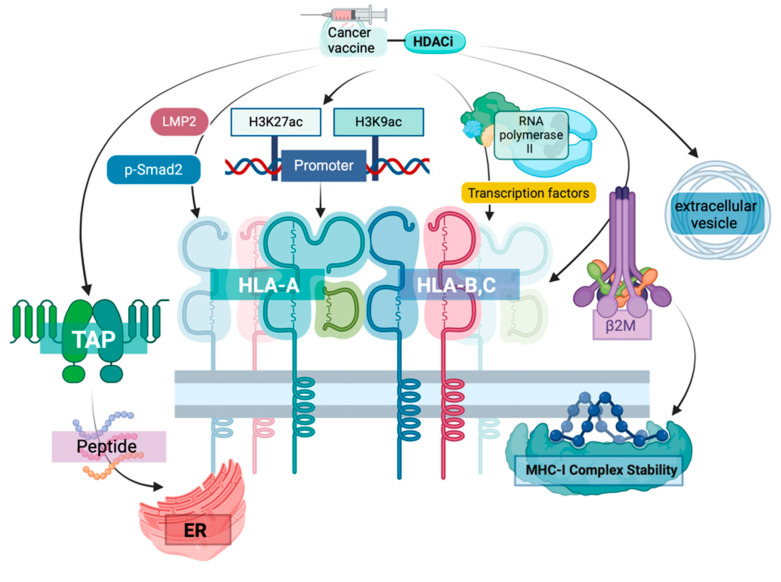
**Mechanisms of HDACi and INT in enhancing MHC-I complex stability and antigen presentation.** HDACi can help INT to improve MHC-I-mediated antigen presentation through multiple pathways. HDACi increases histone acetylation (H3K27ac and H3K9ac) at promoter regions, enhancing transcriptional activation of MHC-I genes. Simultaneously, HDACi upregulates TAP transporter expression through LMP2 and p-Smad2 signaling, facilitating peptide transport into the endoplasmic reticulum (ER). HDACi also promotes RNA polymerase II recruitment and transcription factor activity, leading to increased expression of MHC-I components, including HLA-A, HLA-B, and HLA-C molecules. β2M (β2-Microglobulin) production is significantly upregulated, which is essential for proper MHC-I assembly and stability on the cell surface. Additionally, HDACi modulates the composition of extracellular vesicles, enhancing intercellular transport of MHC-I complexes.

**Figure 3 vaccines-13-00550-f003:**
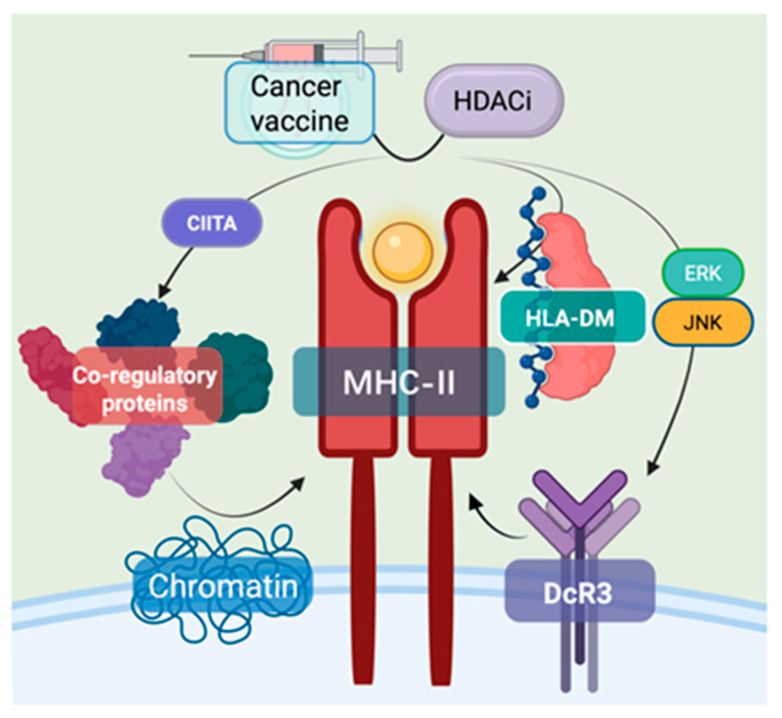
**Synergistic effects of INT and HDACi on MHC class II-mediated antitumor immunity.** HDACi can deliver tumor-associated antigens (TANs) to antigen-presenting cells, therefore enhancing MHC class II expression. This combination activates specific chemokine cascades: CXCL8 activates CXCR1/2 receptors, subsequently stimulating both PI3K/AKT and MAPK pathways, while CXCL9/10/11 directly facilitate immune cell recruitment and aggregation within the tumor microenvironment. The PI3K/AKT signaling axis modulates myeloid-derived suppressor cell (MDSC) activity. This mechanistic synergy between epigenetic modulation and vaccination represents a rational approach for enhancing antigen presentation within the tumor microenvironment, thereby offering a promising strategy for targeted cancer immunotherapy.

**Figure 4 vaccines-13-00550-f004:**
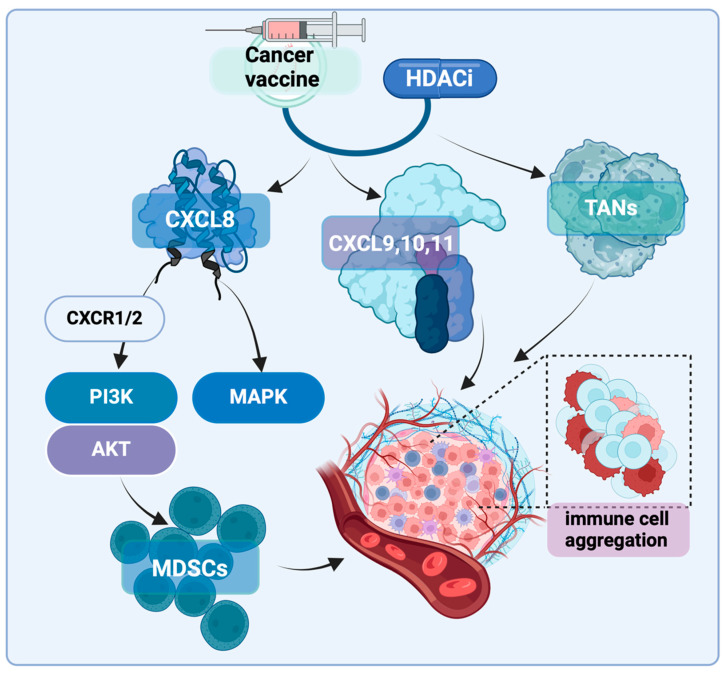
**HDACi enhances mRNA INT efficacy through modulation of chemokine signaling pathways.** HDACi regulates multiple chemokine axes that influence immune cell recruitment and function. HDACi suppresses CXCL8 expression, which normally activates CXCR1/2 receptors to trigger PI3K/AKT and MAPK signaling pathways that promote myeloid-derived suppressor cell (MDSC) recruitment and immunosuppression. Simultaneously, HDACi upregulates CXCL9, CXCL10, and CXCL11 expression, enhancing T cell chemotaxis to the tumor site. HDACi also reprograms tumor-associated neutrophils (TANs) from a pro-tumorigenic to an anti-tumor phenotype.

**Table 1 vaccines-13-00550-t001:** Emerging targets and potential mechanism of HDACi.

Emerging HDACi Targets	Full Name	Potential Mechanism	Cancer Type
LTR12	Long terminal repeat 12	Activate multiple cryptic transcription start sites of LTR12 elements	Prostate cancer, Liver cancer
HERV	Human endogenous retroviruses	Directly influence HERV sites in T cells, and the upregulated ERVs can be recognized by Pattern Recognition Receptors (PRRs)	Colorectal cancer, Triple-negative breast cancer
TAP2	Transporter 2	Upregulate the expression of TAP2	Melanoma
LMP2	Latent Membrane Protein 2	Upregulate the expression of LMP2	Melanoma
LMP7	Latent Membrane Protein 7	Upregulate the expression of LMP7	Melanoma
MHC class I	Major Histocompatibility Complex Class I	Induce STAT1 and Smad2/3 phosphorylation in NSCLC cells, leading to increased MHC class I expression	Melanoma

**Table 2 vaccines-13-00550-t002:** Emerging targets and potential mechanism of HDACi.

Trial Number	Launch	Phase	Study Status	HDACi (Targets)	INT	Other Combined Agents	Cancer Type	Patient Numbers	Endpoints	Preliminary Results
NCT02886065	2017	Ib	Active	Citarinostat	PVX-410	-	Smoldering Multiple Myeloma	19	Safety and Tolerability of the Vaccine	No Results Posted
NCT05898828	2024	I/II	Withdrawn	Entinostat	H1299 cell lysate vaccine	Nivolumab, Montanide(R) ISA-51 VG	Advanced esophageal Cancer	0	Safe Dose/Frequency of Immunologic Responses	No Results Posted
NCT04296942	2021	I	Terminated	Entinostat	BN-Brachyury vaccine	M7824, T-DM1	Advanced Stage Breast Cancer	1	Overall Response	Progression/Recurrence Time is 5 Months and 17 Days

## Data Availability

Not applicable.

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
