# Peer review of "Synergistic Integration of HDAC Inhibitors and Individualized Neoantigen Therapy (INT): A Next-Generation Combinatorial Approach for Cancer Immunotherapy"

_vaccines, 2025, doi:10.3390/vaccines13060550_

Round 1
Reviewer 1 Report
Comments and Suggestions for Authors
The Review article titled 'Synergistic Integration of HDAC Inhibitors and Cancer Vaccines: A Next-Generation Combinatorial Approach for Cancer Immunotherapy' is very interesting. It outlines the role of HDAC inhibitors and cancer vaccines for the treatment of various diseases. The review article could be improved with the following suggestions.
- The term Cancer vaccine needs to be replaced with individualized neoantigen therapy (INT) as the therapy is more tailored towards individualized patients.
- The title also need to be also replaced with INT instead of cancer vaccines.
- The authors need to include a separate section on how the HDAC could be targeted by HDACi and their delivery mechanisms. with more focus on how the HDAC could be downregulated with emerging technologies.
- It is worth mentioning how genome editing mechanisms could be used to downregulate the HDACs using CRISPR-Cas, Prime editing, TALENS etc.
- A table highlighting the emerging targets that could be targeted would gain attention.
Author Response
Dear Editor and Reviewers,
We sincerely thank you for the careful review and constructive comments on our manuscript. We have thoroughly addressed all the comments and suggestions, which have helped us significantly improve the quality of our manuscript. Below, we provide point-by-point responses to each comment, with the corresponding revisions highlighted in the revised manuscript.
Rui Han
The Review article titled 'Synergistic Integration of HDAC Inhibitors and Cancer Vaccines: A Next-Generation Combinatorial Approach for Cancer Immunotherapy' is very interesting. It outlines the role of HDAC inhibitors and cancer vaccines for the treatment of various diseases. The review article could be improved with the following suggestions.
1.The term Cancer vaccine needs to be replaced with individualized neoantigen therapy (INT) as the therapy is more tailored towards individualized patients.
Answer:
We thank our reviewer for his/her valuable suggestions. We have changed the term "cancer vaccine" to "Individualized neoantigen therapy (INT)", and the changed part has been marked in yellow in the text.
2.The title also need to be also replaced with INT instead of cancer vaccines.
Answer:
Many thanks to our reviewer for such meticulous reviewing .The relevant part in the title has been changed to INT(Line 1-4).
3.The authors need to include a separate section on how the HDAC could be targeted by HDACi and their delivery mechanisms. with more focus on how the HDAC could be downregulated with emerging technologies.
Answer:
We greatly appreciate our reviewer for their meticulous and thorough review. We totally agree that methos of genome editing in HDACs downregulation is important. Therefore, we added a whole new section titled as “3. Downregulation of HDACs Using Genome Editing Mechanisms” to introduce such area. We have summarized various novel technologies: CRISPR-Cas9, dCas9, Prime Editing, TALEN, RNA interference technology and CRISPR-Cas13 systems, as well as switchable Cas12a-based systems. These new additions can further enrich our manuscript's discussion on HDAC regulation approaches, providing more technical insights for subsequent research on the role of HDACs in cancer immunotherapy, and contributing to the development of new cancer treatment strategies based on HDAC regulation. The modified content has been highlighted in yellow in the manuscript (Line 559-661) and also has been shown as followed:
“3. Downregulation of HDACs Using Genome Editing Mechanisms
The evolution of genome editing technologies has revolutionized our approach to HDAC downregulation, offering unprecedented precision and versatility. DNA-targeting platforms such as CRISPR-Cas9, Prime Editing, and TALENs, alongside RNA-targeting systems like CRISPR-Cas13, each present distinct advantages tailored to specific research objectives[108]. Moreover, deactivated variants (dCas9 and dCas13) enable reversible regulation without permanent genomic alterations, proving invaluable for investigating HDAC functions and developing therapeutic strategies[109]. Moving forward, research priorities should include enhancing specificity and efficiency, minimizing off-target effects, and creating more sophisticated delivery systems to facilitate clinical translation. Integrating bioinformatics with high-throughput screening methodologies to identify optimal targets will enable precise HDAC expression control, potentially transforming treatment approaches for epigenetic disorders and cancer.
3.1 CRISPR-Cas9 Mediated Gene Editing
CRISPR-Cas9 (Clustered Regularly Interspaced Short Palindromic Repeats and CRISPR-associated protein 9) represents a revolutionary breakthrough in genome editing technology. The mechanism begins with transcription of the CRISPR region to generate crRNA, which then combines with tracrRNA to form single guide RNA (sgRNA)[110]. This complex guides the Cas9 endonuclease to specific genomic sequences, creating double-strand breaks (DSBs) in the DNA. Cellular repair mechanisms address these breaks through either homology-directed repair (HDR) or non-homologous end joining (NHEJ)[111].
To effectively downregulate HDAC genes, researchers can design sgRNAs targeting coding regions or regulatory elements, directing Cas9 to cleave the DNA at these locations. When NHEJ repair occurs, it frequently introduces frameshift mutations, deletions, or insertions that compromise gene function or substantially reduce expression[112]. Evidence have demonstrated that successfully disrupting specific HDAC gene regions using CRISPR-Cas9 not only diminishes expression of the targeted HDAC but also influences associated signaling networks and cellular functions[113].
Interestingly, HDACi can enhance the efficiency of advanced CRISPR-Cas9 sys-tems. Research indicates that inhibiting HDAC1 and HDAC2 significantly improves both knockout and knock-in efficiency. This enhancement occurs because HDACi treatment relaxes chromatin structure, creating more opportunities for Cas9 to access target DNA. Specifically, compounds like Entinostat and Panobinostat increase H3 histone acetylation in chromatin, amplifying CRISPR-Cas9 editing capabilities[113].
3.2 Transcriptional Repression Mediated by Deactivated Cas9 (dCas9)
Deactivated Cas9 (dCas9), engineered to lack nuclease activity while retaining DNA-binding capacity, offers an elegant approach to HDAC regulation. When guided to HDAC gene promoters or transcription initiation sites, dCas9 can dramatically re-duce expression by physically blocking RNA polymerase, ribosome binding, or transcription factor interactions[114]. This technique, termed CRISPR interference (CRIS-PRi), avoids DNA breaks, thereby circumventing potential off-target complications and chromosomal rearrangements associated with traditional CRISPR-Cas9 applications[115].
Researchers have successfully employed CRISPRi to target critical regulatory elements within HDAC promoter regions, effectively suppressing specific HDAC sub-types[116]. Furthermore, the fusion of transcriptional repression domains (particularly KRAB) to dCas9 substantially enhances inhibitory effects, yielding more pronounced reduction in HDAC expression levels. This approach shows remarkable promise for therapeutic applications[117].
3.3 Prime Editing Technology
Prime Editing emerges as a sophisticated genome editing platform offering novel strategies for HDAC downregulation. This technology ingeniously combines modified Cas9 with reverse transcriptase, guided by specialized pegRNA (prime editing guide RNA), enabling precise nucleotide substitutions, insertions, or deletions without generating double-strand breaks[118]. For manipulating HDAC genes, Prime Editing can introduce specific mutations in functional domains or regulatory elements—such as premature stop codons, splice site alterations, or promoter mutations—affecting transcription, translation, or splicing to achieve targeted downregulation[119]. Evidence suggests that introducing specific mutations in HDAC gene promoter regions can disrupt transcription factor binding sites, inhibiting gene expression and ultimately reducing protein levels[120]. The exceptional precision and reduced off-target effects position Prime Editing as particularly promising for clinical applications.
3.4 TALEN Technology
TALEN (Transcription Activator-Like Effector Nuclease) technology comprises three key components: an N-terminal domain containing nuclear localization signals, a central domain with TALE repeat sequences recognizing specific DNA motifs, and a C-terminal domain featuring FokI nuclease activity[121]. By designing TALENs specifically targeting HDAC genes, researchers can direct FokI to create double-strand breaks at precise locations[121].
When cellular repair mechanisms address these breaks, mutations or deletions often result, effectively downregulating HDAC expression. Compared to CRISPR-Cas9, TALENs offer enhanced specificity and fewer off-target effects, despite their more complex design requirements[122]. Studies have documented successful TALEN-mediated downregulation of various HDAC subtypes, with corresponding epigenetic modifications and biological consequences.
3.5 RNA Interference Technology and CRISPR-Cas13 System
RNA interference (RNAi) directly targets HDAC transcripts, inhibiting protein expression by specifically binding and degrading mRNA through short interfering RNA (siRNA) or short hairpin RNA (shRNA)[115]. Evidence has confirmed that siR-NAs or shRNAs targeting specific HDAC subtypes effectively reduce corresponding expression levels and alter critical cellular processes including proliferation, differentiation, and apoptosis[123].
The CRISPR-Cas13 system represents an innovative RNA-targeting approach for HDAC downregulation. Unlike DNA-targeting CRISPR-Cas9, Cas13 specifically targets RNA molecules[124]. Researchers can design guide RNAs targeting HDAC transcripts, enabling Cas13 to bind and degrade these mRNAs, thereby reducing HDAC protein expression[124]. Recent advances have introduced deactivated Cas13 (dCas13) for translational-level CRISPR interference (Tl-CRISPRi)[125]. This methodology doesn't cause mRNA degradation but instead prevents protein synthesis by blocking translation. Optimized guide RNAs direct dCas13 to HDAC mRNA translation initiation regions, preventing ribosome binding or elongation, thereby precisely controlling HDAC protein levels. This technology has successfully achieved simultaneous downregulation of multiple targets, demonstrating exceptional specificity and versatility[126].
3.6 Switchable Cas12a-Based System
An innovative switchable system based on Cas12a presents a novel tool for HDAC activity monitoring and regulation. By incorporating specific lysine acetylation modifications to Cas12a protein, researchers can initially suppress its nuclease activity; when HDACs remove these acetylation marks, Cas12a function is restored[127]. This characteristic has been cleverly adapted to develop sensitive HDAC activity assays while providing new approaches for targeted HDAC downregulation[128]. Theoretically, conditional Cas12a systems could be engineered to activate in specific cellular environments, downregulating target HDACs only under predetermined conditions.”
4.It is worth mentioning how genome editing mechanisms could be used to downregulate the HDACs using CRISPR-Cas, Prime editing, TALENS etc.
Answer:
We thank our reviewer for his/her professional suggestion. We totally agree that methos of genome editing in HDACs downregulation is important. Therefore, we added a whole new section titled as “3. Downregulation of HDACs Using Genome Editing Mechanisms” to introduce such area. We have summarized various novel technologies: CRISPR-Cas9, dCas9, Prime Editing, TALEN, RNA interference technology and CRISPR-Cas13 systems, as well as switchable Cas12a-based systems. These new additions can further enrich our manuscript's discussion on HDAC regulation approaches, providing more technical insights for subsequent research on the role of HDACs in cancer immunotherapy, and contributing to the development of new cancer treatment strategies based on HDAC regulation. The modified content has been highlighted in yellow in the manuscript (Line 559-661) and has already been shown above (same as answer 3).
5.A table highlighting the emerging targets that could be targeted would gain attention.
Answer:
Many thanks to our reviewer for his/her rigorous and meticulous review. We fully agree with the reviewer's opinion on adding a table to highlight the potential targets of HDACi. The new table has been made as shown below. It has been marked yellow in the main text and is located at(Line 557-558)
Table1 Emerging targets and potential mechanism of HDACi
|
Emerging HDACi targets |
Full name |
Potential mechanism |
Cancer type |
|
LTR12 |
Long terminal repeat 12 |
Activate multiple cryptic transcription start sites of LTR12 elements |
Prostate cancer,Liver cancer |
|
HERV |
Human endogenous retroviruses |
Directly influence HERV sites in T cells, and the upregulated ERVs can be recognized by Pattern Recognition Receptors (PRRs) |
Colorectal cancer,Triple - negative breast cancer |
|
TAP2 |
Transporter 2 |
Upregulate the expression of TAP2 |
Melanoma |
|
LMP2 |
Latent Membrane Protein 2 |
Upregulate the expression of LMP2 |
Melanoma |
|
LMP7 |
Latent Membrane Protein 7 |
Upregulate the expression of LMP7 |
Melanoma |
|
MHC class I |
Major Histocompatibility Complex Class I |
Induce STAT1 and Smad2/3 phosphorylation in NSCLC cells, leading to increased MHC class I expression |
Melanoma |

Reviewer 2 Report
Comments and Suggestions for Authors
The manuscript titled "Synergistic Integration of HDAC Inhibitors and Cancer Vaccines: A Next‑Generation Combinatorial Approach for Cancer Immunotherapy" by Rui Han et al. presents a timely and in‑depth review of combining histone deacetylase inhibitors (HDACi) with cancer vaccines, detailing mechanistic rationales and early clinical experiences. The topic is important and the breadth of coverage — from endogenous retroelement activation to chemokine modulation — is impressive. However, several conceptual and editorial improvements are needed to enhance the manuscript's readability and scientific rigor.
Overall, the manuscript is comprehensive, but its impact is diminished by a few editorial oversights. Addressing the following points will ensure publication in the journal.
- The introduction outlines the landscape of cancer vaccines and HDACi monotherapy but does not clearly position this combinatorial strategy within the context of other emerging adjuvant approaches (e.g., TLR agonists, STING agonists). How does the HDACi–vaccine pairing compare in potency and safety to these other adjuvant combinations in preclinical or early‑phase trials? Please, add this information.
- Table 1 lists trial identifiers, but key details (patient numbers, endpoints, preliminary results) are missing, making it hard to gauge clinical maturity. Can the authors provide even brief summaries of available safety/efficacy readouts in the text (e.g., response rates, biomarker changes) for the trials in Table 1? Also, in my opinion, the last column (Level of evidence) is unnecessary.
- There are some grammatical and editorial corrections:
(Line 107) Original: “HDAC inhibitors (HDACi) , on the other hand, are compounds…”
Correct: “HDAC inhibitors (HDACi), on the other hand, are compounds…”
(Line 361) Original: “Evidence has fund that HDACi can enhance peptide transport…”
Correct: “Evidence has found that HDACi can enhance peptide transport…”
Original: “(Figure.1), (Figure.2), (Figure.3), (Figure.4)”
Correct: “(Figure 1) and etc.”
Put a space before each reference: “… to enhance anti-tumor responses[1].”
Conclusion:
This review covers an exciting frontier in cancer immunotherapy. Correcting the few editorial slips and adding new information will significantly enhance its clarity and impact.
Author Response
Dear Editor and Reviewers,
We sincerely thank you for the careful review and constructive comments on our manuscript. We have thoroughly addressed all the comments and suggestions, which have helped us significantly improve the quality of our manuscript. Below, we provide point-by-point responses to each comment, with the corresponding revisions highlighted in the revised manuscript.
Rui Han
The manuscript titled "Synergistic Integration of HDAC Inhibitors and Cancer Vaccines: A Next Generation Combinatorial Approach for Cancer Immunotherapy" by Rui Han et al. presents a timely and in depth review of combining histone deacetylase inhibitors (HDACi) with cancer vaccines, detailing mechanistic rationales and early clinical experiences. The topic is important and the breadth of coverage — from endogenous retroelement activation to chemokine modulation — is impressive. However, several conceptual and editorial improvements are needed to enhance the manuscript's readability and scientific rigor.
Overall, the manuscript is comprehensive, but its impact is diminished by a few editorial oversights. Addressing the following points will ensure publication in the journal.
1.The introduction outlines the landscape of cancer vaccines and HDACi monotherapy but does not clearly position this combinatorial strategy within the context of other emerging adjuvant approaches (e.g., TLR agonists, STING agonists). How does the HDACi–vaccine pairing compare in potency and safety to these other adjuvant combinations in preclinical or early phase trials? Please, add this information.
Answer:
Our team greatly appreciate the insightful comment regarding the positioning of our HDACi-vaccine strategy within the broader context of adjuvant approaches of our reviewer. We have re-written and improved our “Introduction” by providing a more comprehensive paragraph that introduces other major adjuvant strategies including TLR agonists (imiquimod, CpG oligodeoxynucleotides), STING agonists, and cytokines (GM-CSF, IL-12), outlining their mechanisms of action and limitations. We have also explicitly positioned HDACi relative to these alternatives, highlighting their unique advantages through simultaneous modification of the tumor epigenetic landscape and immune microenvironment, while emphasizing their established safety profiles from previously approved indications. This addition provides the contextual framework you requested, allowing readers to understand how our proposed HDACi-vaccine combination compares with and potentially offers advantages over other adjuvant strategies in development. The revised introduction has been marked in yellow (Line 39-88), and shown as below:
“Therapeutic cancer vaccines represent a promising immunotherapeutic approach that stimulates the immune system to recognize and eliminate cancer cells. Individualized neoantigen therapy(INT) as a special type of cancer vaccine has demonstrated certain advantages in clinical practice[1]. The recent success of the KEYNOTE-942 trial, which demonstrated significant improvement in recurrence-free survival for melanoma patients receiving an mRNA-based INT combined with pembrolizumab, has renewed interest in this modality. This trial showed a 44% reduction in risk of recurrence or death compared to pembrolizumab monotherapy, with a recurrence-free survival rate of 78.6% versus 62.2% at 18 months median follow-up[2].
Despite their potential, INT face several challenges that limit their efficacy as monotherapy. These include tumor heterogeneity, immune evasion mechanisms, immunosuppressive microenvironments, and insufficient presentation of tu-mor-associated antigens[3]. The tumor microenvironment can suppress vac-cine-induced immune responses through regulatory T cells, myeloid-derived suppressor cells, and inhibitory immune checkpoints[4]. Additionally, downregulation of antigen presentation machinery in tumor cells further compromises vaccine effectiveness. These challenges highlight the critical need for adjuvant strategies to enhance INT potency[4].
Several adjuvant approaches have emerged to address these limitations. Toll-like receptor (TLR) agonists like imiquimod and CpG oligodeoxynucleotides enhance innate immune activation but can cause inflammatory side effects[5]. STING agonists demonstrate potent type I interferon induction and tumor regression in preclinical models, though they may trigger autoimmune-like syndromes at higher doses[6]. Cytokines such as GM-CSF and IL-12 boost T-cell responses but have narrow therapeutic windows[7]. When compared with these approaches, histone deacetylase inhibitors (HDACi) potentially offer unique advantages through their ability to simultaneously modify the tumor epigenetic landscape and immune microenvironment with established safety profiles from their approved indications[8].
Histone deacetylases (HDACs) are enzymes that remove acetyl groups from lysine residues on histones and are classified into four major families: Class I (HDAC1, 2, 3, 8), Class II (HDAC4-7, 9-10), Class III (sirtuins, SIRT1-7), and Class IV (HDAC11)[9, 10]. HDAC inhibitors (HDACi) increase chromatin accessibility by inhibiting histone deacetylation, thereby altering gene expression patterns[9, 10]. Although the FDA has approved several HDACi for hematological malignancies, such as vorinostat for cutaneous T-cell lymphoma, clinical trials of monotherapy in solid tumors have generally yielded unfavorable results[11, 12]. Current research primarily focuses on combination strategies, including HDACi with immune checkpoint inhibitors, targeted therapies, or cellular therapies like CAR-T[13].
Our investigations into HDACi's immunomodulatory effects have revealed their capacity to enhance tumor antigen presentation, promote effector T cell infiltration, and inhibit myeloid-derived suppressor cell function[8]. Despite limited efficacy as monotherapy, HDACi show particular promise as combination partners with INT due to their ability to overcome tumor heterogeneity and immune evasion mechanisms[14-16]. The complementary actions of these two modalities—with cancer vaccines providing specific tumor antigens and HDACi enhancing their presentation and recognition—create a compelling rationale for combination therapy.
This review provides the first comprehensive analysis of the mechanistic rationale for integrating HDACi with INT as a next-generation combinatorial approach for cancer immunotherapy. We examine the synergistic effects, early clinical evidence, potential challenges, and future directions of this promising strategy that could trans-form treatment paradigms for patients with limited therapeutic options.”
2.Table 1 lists trial identifiers, but key details (patient numbers, endpoints, preliminary results) are missing, making it hard to gauge clinical maturity. Can the authors provide even brief summaries of available safety/efficacy readouts in the text (e.g., response rates, biomarker changes) for the trials in Table 1? Also, in my opinion, the last column (Level of evidence) is unnecessary.
Answer:
We extend our heartfelt gratitude to the reviewer for his/her meticulous and thorough reviewing for Table 2(Because a table was added in front, the original Table 1 became Table 2). We have supplemented and improved the key details of the trial (such as patient numbers, endpoints, preliminary results) as required. The relevant parts have been marked in yellow in the manuscript(Line 674-675)
Table 2. Completed and ongoing clinical trials of HDACi in combination with tumor vaccines or other adjuvants.
|
Trial Number |
Launch |
Phase |
Study Status |
HDACi (targets) |
INT |
Other combined agents |
Cancer type |
Patient numbers |
Endpoints |
Preliminary results |
|
|
NCT02886065 |
2017 |
Ib |
Active |
Citarinostat
|
PVX-410 |
- |
Smoldering Multiple Myeloma |
19 |
Safety And Tolerability Of The Vaccine |
No Results Posted |
|
|
NCT05898828 |
2024 |
I/II |
Withdrawn |
Entinostat |
H1299 cell lysate vaccine |
Nivolumab, Montanide(R) ISA-51 VG |
Advanced esophageal Cancer |
0 |
Safe dose/frequency of immunologic responses |
No Results Posted |
|
|
NCT04296942 |
2021 |
I |
Terminated |
Entinostat |
BN-Brachyury vaccine |
M7824,T-DM1 |
Advanced Stage Breast Cancer |
1 |
Overall Response |
Progression/recurrentce time is 5 months and 17 days |
|
3.There are some grammatical and editorial corrections:
(Line 107) Original: “HDAC inhibitors (HDACi) , on the other hand, are compounds…”
Correct: “HDAC inhibitors (HDACi), on the other hand, are compounds…”
Answer:
Many thanks to the reviewer for his/her meticulous reviewing efforts and attention to detail. This part was originally in the Introduction. Now this part has been fully refined and rewritten(Line 38-88). Meanwhile, we have checked all the relevant grammar issues in the article to ensure that no related problems are expressed.
(Line 361) Original: “Evidence has fund that HDACi can enhance peptide transport…”
Correct: “Evidence has found that HDACi can enhance peptide transport…”
Answer:
We are extremely thankful to our reviewer for such a meticulous review. We have corrected the incorrect spelling of the words (Line 363)in this sentence and checked other spelling problems to ensure the correct spelling.
Original: “(Figure.1), (Figure.2), (Figure.3), (Figure.4)”
Correct: “(Figure 1) and etc.”
Put a space before each reference: “… to enhance anti-tumor responses[1].”
Answer:
Our sincere thanks are due to the reviewer for his/her dedicated reviewing work. We have adjust the format into correct version (Line 151, 162, 194, 210, 238, 354, 375, 418, 442, 463, 515, 534, 551, 672, 691, 694).
We further reviewed the full text and corrected the improved format issues mentioned above.
Reviewer 3 Report
Comments and Suggestions for Authors
This is a comprehensive review on synergistic mechanisms of HDAC inhibitors and Cancer Vaccines. The manuscript was well-organized, presenting a clear and coherent narrative that contributes meaningfully to the cancer immunotherapy field. The manuscript aligns well with the journal’s scope and standards.
Specific comments:
- While the manuscript is generally clear, there are places that could benefit from additional information to enhance clarity. For example, beginning with administration, a vaccine must undergo several critical and complex steps to achieve effectiveness. Depending on the specific pathways, vaccines can target and therefore act on antigen presenting cells, effector cells, or tumor cells. The majority of the discussion of the paper was focused on HDACi’s impact on tumor cells (except 2.1 there were discussions on T cells). Since DCs are import cells in vaccine, it will greatly improve the manuscript if the authors also discuss the coordination mechanism of HDACi on DCs. The antigen processing and presentation are different in tumor cells and in professional antigen presenting cells. This also significantly influences the corresponding delivery approach (4.7).
- The reference list is comprehensive. Nonetheless, in some sections a substantial portion of the discussion lacks citation of supporting references (for example, line 212-218).
- In figure 3. MHC class II molecules should have two membrane anchors instead of one. Unlike MHC I, MHC II both chains are transmembrane proteins, meaning each one spans the membrane and has its own membrane anchor domain.
Author Response
Dear Editor and Reviewers,
We sincerely thank you for the careful review and constructive comments on our manuscript. We have thoroughly addressed all the comments and suggestions, which have helped us significantly improve the quality of our manuscript. Below, we provide point-by-point responses to each comment, with the corresponding revisions highlighted in the revised manuscript.
Rui Han
This is a comprehensive review on synergistic mechanisms of HDAC inhibitors and Cancer Vaccines. The manuscript was well-organized, presenting a clear and coherent narrative that contributes meaningfully to the cancer immunotherapy field. The manuscript aligns well with the journal’s scope and standards.
Specific comments:
1.While the manuscript is generally clear, there are places that could benefit from additional information to enhance clarity. For example, beginning with administration, a vaccine must undergo several critical and complex steps to achieve effectiveness. Depending on the specific pathways, vaccines can target and therefore act on antigen presenting cells, effector cells, or tumor cells. The majority of the discussion of the paper was focused on HDACi’s impact on tumor cells (except 2.1 there were discussions on T cells). Since DCs are import cells in vaccine, it will greatly improve the manuscript if the authors also discuss the coordination mechanism of HDACi on DCs. The antigen processing and presentation are different in tumor cells and in professional antigen presenting cells. This also significantly influences the corresponding delivery approach (4.7).
Answer:
We thank our reviewer for his/her valuable comments. The indication of the importance of dendritic cells (DCs) in cancer vaccine research and the suggestion to explore the coordination mechanism of histone deacetylase inhibitors (HDACi) on DCs are, indeed, of great significance for improving the quality of the manuscript. we have now added a section of “Synergizing with DCs” to discuss the detailed elaboration on the effects of HDACi on DCs in the use of INT. This section deeply explored the impact of HDACi on multiple biological functions of DCs, including promoting DC maturation, bidirectionally regulating antigen uptake, enhancing antigen processing and presentation capabilities, altering the cytokine secretion profile, and regulating migration ability. We also introduced the latest research findings on the indirect influence of HDACi on DC function through the regulation of microRNAs, and related clinical research progress, etc . Related content has been added at Line 90-139(marked in yellow), and has been shown as followed:
“2.1 Synergizing with DCs
Dendritic cells (DCs), as professional antigen-presenting cells, play a central role in cancer immunotherapy, particularly in the process of INT activating the immune system[17]. HDACi exerts complex and multifaceted regulatory effects on DCs with significant dose and time dependency, demonstrating powerful synergistic potential.
Studies indicate that HDACi significantly influences multiple biological functions of DCs and promotes DC maturation. HDACi inhibits HDAC activity, increases acety-lation levels of histones H3 and H4, loosens chromatin structure, thereby facilitating the binding of DC maturation-related transcription factors such as NF-κB, AP-1, and STAT1 to target gene promoter regions, upregulating the expression of CD80, CD86, CD40, and MHC molecules[18]. Notably, different types of HDACi affect DC maturation differently; SAHA (Vorinostat) and TSA (Trichostatin A) primarily promote DC maturation through inhibition of HDAC1 and HDAC2, while MS-275 (Entinostat) tends to selectively inhibit HDAC1[19-21]. Regarding antigen uptake and processing, recent research reveals that HDACi can bidirectionally regulate DCs' antigen uptake capacity, closely related to HDACi's temporal window of action[20]. HDACi interven-tion enhances cytoskeletal dynamics and regulates endocytic receptors (such as Man-nose receptors and Scavenger receptors), promoting tumor antigen uptake by DCs; however, prolonged treatment may reduce uptake efficiency[18]. In the antigen processing stage, HDACi enhances antigen cleavage and transport efficiency by upregulating immunoproteasome subunits (such as LMP2, LMP7, and MECL-1) and TAP1/2 transport proteins[22](Table 1). Particularly, HDACi promotes the activity of lysosomal proteins like Cathepsin S and Cathepsin B, optimizing the MHC class II molecule antigen processing pathway[23]. During antigen presentation, HDACi not only upregulates MHC class II molecules by increasing CIITA (MHC class II transcriptional acti-vator) expression but also expands DCs' capacity to present lipid antigens by enhancing expression of antigen presentation-associated auxiliary molecules like CD1d, activating broader T cell subsets, including NKT cells[23]. Additionally, HDACi significantly alters DCs' cytokine secretion profile, promoting production of pro-inflammatory factors like IL-12p70, IL-15, and IFN-α, while inhibiting the release of immunosuppressive factors such as IL-10 and TGF-β, creating a microenvironment more conducive to T cell activation[24]. HDACi also enhances DCs' ability to migrate to lymph nodes by regulating chemokine receptors like CCR7 and CXCR4, crucial for effectively initiating anti-tumor immune responses[25]. New discoveries in trans-epigenetics reveal that HDACi can indirectly influence DC function by regulating microRNAs (such as miR-155 and miR-146a) expression[26].
In clinical translation research, sequential treatment strategies combining HDACi with DC vaccines (pretreatment of tumor microenvironment with HDACi followed by DC vaccine inoculation) have demonstrated significant synergistic anti-tumor effects in melanoma, pancreatic cancer, and non-small cell lung cancer models, not only enhancing CD8+ T cell infiltration and activity but also reducing the proportion of regulatory T cells (Tregs) and myeloid-derived suppressor cells (MDSCs) in the tumor microenvironment[27]. Notably, in a Phase I/II clinical trial reported in 2023, the combi-nation of Panobinostat with autologous DC vaccines for recurrent glioblastoma patients achieved preliminary positive results, with 30% of patients achieving disease stability or partial remission, and treatment group patients showing marked expansion of tumor-specific CD8+ T cell populations[28]. Nevertheless, HDACi regulation of DC function still faces challenges including narrow dosage windows, insufficient selectivity, and clinical medication sequencing. Future efforts should focus on developing more selective HDACi subtype-specific inhibitors and optimizing combination strategies with immunotherapies to maximize their potential in cancer immunotherapy.”
2.The reference list is comprehensive. Nonetheless, in some sections a substantial portion of the discussion lacks citation of supporting references (for example, line 212-218).
Answer:
We are extremely thankful to our reviewer for such a meticulous review. The references in lines 212 to 218 have been completed. The added references have been marked in yellow in the text(Line 220).
3.In figure 3. MHC class II molecules should have two membrane anchors instead of one. Unlike MHC I, MHC II both chains are transmembrane proteins, meaning each one spans the membrane and has its own membrane anchor domain.
Answer:
Many thanks to our reviewer for such meticulous reviewing. We have revised the symbol of MHC class II in figure 3 and attached it as following:
Reviewer 4 Report
Comments and Suggestions for Authors
The present manuscript (ID: vaccines-3599073) titled "Synergistic Integration of HDAC Inhibitors and Cancer Vaccines: A Next-Generation Combinatorial Approach for Cancer Immunotherapy" shows that Combining HDAC inhibitors with cancer vaccines could make the immune system fight cancer more effectively. This new approach looks very promising, but more information is needed to make sure it is safe and works well in patients.
Abstract. It is well-structured, introduces the main idea clearly. However, key points (like what HDAC inhibitors do, and how they help vaccines) could have been highlighted more sharply without getting mixed with too much background.
Introduction. The Introduction repeats similar ideas about the challenges of cancer vaccines. It mentions the limitations of cancer vaccines (like poor immune activation) more than once, in slightly different ways. This repetition makes the Introduction longer than needed and slows down the movement toward the main point (the combination with HDAC inhibitors). Combine or tighten these points into one strong paragraph. The main topic (HDAC inhibitors helping cancer vaccines) only appears clearly later in the Introduction.
Potential Coordination Mechanisms. The authors use a lot of specialized terms like "epigenetic reprogramming," "neoantigen presentation," "cGAS-STING pathway," and "class I HDAC inhibition" one after another. Short explanations will be provided after each technical term. Simple examples or small diagrams should be provided to make the complex ideas easier to understand.
Clinical Applications of HDACi Combined with Tumor Vaccines. This sections talks about early clinical trials combining HDAC inhibitors with vaccines (PVX-410, H1299 vaccine). However, the presence of limited clinical data makes it feel a bit speculative.
Current Issues and Challenges. This part discusses challenges like PD-L1 expression, safety issues, optimal drug sequencing, and need for biomarkers. Some sections are slightly repetitive (especially about PD-L1) that should be improved.
Author Response
Dear Editor and Reviewers,
We sincerely thank you for the careful review and constructive comments on our manuscript. We have thoroughly addressed all the comments and suggestions, which have helped us significantly improve the quality of our manuscript. Below, we provide point-by-point responses to each comment, with the corresponding revisions highlighted in the revised manuscript.
Rui Han
The present manuscript (ID: vaccines-3599073) titled "Synergistic Integration of HDAC Inhibitors and Cancer Vaccines: A Next-Generation Combinatorial Approach for Cancer Immunotherapy" shows that Combining HDAC inhibitors with cancer vaccines could make the immune system fight cancer more effectively. This new approach looks very promising, but more information is needed to make sure it is safe and works well in patients.
Abstract. It is well-structured, introduces the main idea clearly. However, key points (like what HDAC inhibitors do, and how they help vaccines) could have been highlighted more sharply without getting mixed with too much background.
Answer:
Many thanks to the reviewer for his/her meticulous reviewing efforts and attention to detail. We fully agree with the reviewers' comments, such as emphasizing the role of HDAC inhibitors in this section and how they help vaccines, and reducing the description of the background. We have rewritten the abstract as required and marked it in yellow(Line 17-34).
Abstract
Background/Objectives: Cancer immunotherapy has advanced, yet therapeutic resistance and low response rates remain problematic. This study explores histone deacetylase inhibitors (HDACi) as adjuvants for cancer vaccines to enhance anti-tumor immunity and overcome these challenges.
Methods: A comprehensive review of relevant literature was conducted. Studies on the immunomodulatory mechanisms of HDACi, their effects on Individualized neoantigen therapy (INT), and clinical applications were analyzed. Results: HDACi enhance anti - tumor immunity through multiple mechanisms. They activate endogenous retroelements, expanding the "antigen repository". HDACi also upregulate MHC class I and II molecules, enhance the antigen processing machinery, improve MHC - I complex stability, and remodel the tumor immune microenvironment. Early clinical trials of HDACi combined with peptide vaccines show promising safety and immunological responses. However, challenges exist, such as HDACi - mediated PD-L1 regulation, optimal sequencing strategies, and biomarker development. Conclusions: The combination of HDACi and cancer vaccines has significant potential in cancer immunotherapy. Despite challenges, it offers a new approach to overcome tumor heterogeneity and immune evasion, especially for patients with limited treatment options. Further research on toxicity management, triple - drug combinations, biomarker identification, and delivery systems is needed to fully realize its clinical benefits.
Introduction. The Introduction repeats similar ideas about the challenges of cancer vaccines. It mentions the limitations of cancer vaccines (like poor immune activation) more than once, in slightly different ways. This repetition makes the Introduction longer than needed and slows down the movement toward the main point (the combination with HDAC inhibitors). Combine or tighten these points into one strong paragraph. The main topic (HDAC inhibitors helping cancer vaccines) only appears clearly later in the Introduction.
Answer:
Thank our reviewer for such astute observation regarding the organizational structure and focus of our introduction. We have implemented substantial revisions to address these concerns by consolidating all information about cancer vaccine limitations into a single, focused paragraph and eliminating redundant descriptions of various vaccine types. We have streamlined the flow by introducing HDACi earlier in the introduction and created a clearer logical progression that moves from cancer vaccines and their promise, to their limitations necessitating adjuvant approaches, to comparison with other adjuvant strategies, followed by introduction of HDACi properties and the rationale for HDACi-vaccine combinations. This restructuring has significantly reduced the overall length while maintaining all critical information and ensuring the main topic of HDACi as adjuvants for cancer vaccines is presented more prominently and earlier in the text. In addition, to the best of our knowledge, studies related to this treatment strategy have not been published yet. Therefore, the HDACI-vaccine combination is currently an extremely cutting-edge study internationally. As a result, there are no mature and relevant clinical outcome reports at present, making it difficult to compare its efficacy and safety with other combinations at present. However, the purpose of this work is precisely to promote the assessment of the efficacy and safety of the program by emphasizing the innovativeness of the study. The rewritten and improved “Introduction” has been marked in yellow (Line 38-88), and shown as below:
“Therapeutic cancer vaccines represent a promising immunotherapeutic approach that stimulates the immune system to recognize and eliminate cancer cells. Individualized neoantigen therapy(INT) as a special type of cancer vaccine has demonstrated certain advantages in clinical practice[1]. The recent success of the KEYNOTE-942 trial, which demonstrated significant improvement in recurrence-free survival for melanoma patients receiving an mRNA-based INT combined with pembrolizumab, has renewed interest in this modality. This trial showed a 44% reduction in risk of recurrence or death compared to pembrolizumab monotherapy, with a recurrence-free survival rate of 78.6% versus 62.2% at 18 months median follow-up[2].
Despite their potential, INT face several challenges that limit their efficacy as monotherapy. These include tumor heterogeneity, immune evasion mechanisms, immunesuppressive microenvironments, and insufficient presentation of tumor-associated antigens[3]. The tumor microenvironment can suppress vac-cine-induced immune responses through regulatory T cells, myeloid-derived suppressor cells, and inhibitory immune checkpoints[4]. Additionally, downregulation of antigen presentation machinery in tumor cells further compromises vaccine effectiveness. These challenges highlight the critical need for adjuvant strategies to enhance INT po-tency[4].
Several adjuvant approaches have emerged to address these limitations. Toll-like receptor (TLR) agonists like imiquimod and CpG oligodeoxynucleotides enhance innate immune activation but can cause inflammatory side effects[5]. STING agonists demonstrate potent type I interferon induction and tumor regression in preclinical models, though they may trigger autoimmune-like syndromes at higher doses[6]. Cytokines such as GM-CSF and IL-12 boost T-cell responses but have narrow therapeutic windows[7]. When compared with these approaches, histone deacetylase inhibitors (HDACi) potentially offer unique advantages through their ability to simultaneously modify the tumor epigenetic landscape and immune microenvironment with established safety profiles from their approved indications[8].
Histone deacetylases (HDACs) are enzymes that remove acetyl groups from lysine residues on histones and are classified into four major families: Class I (HDAC1, 2, 3, 8), Class II (HDAC4-7, 9-10), Class III (sirtuins, SIRT1-7), and Class IV (HDAC11)[9, 10]. HDAC inhibitors (HDACi) increase chromatin accessibility by inhibiting histone deacetylation, thereby altering gene expression patterns[9, 10]. Although the FDA has approved several HDACi for hematological malignancies, such as vorinostat for cutaneous T-cell lymphoma, clinical trials of monotherapy in solid tumors have generally yielded unfavorable results[11, 12]. Current research primarily focuses on combination strategies, including HDACi with immune checkpoint inhibitors, targeted therapies, or cellular therapies like CAR-T[13].
Our investigations into HDACi's immunomodulatory effects have revealed their capacity to enhance tumor antigen presentation, promote effector T cell infiltration, and inhibit myeloid-derived suppressor cell function[8]. Despite limited efficacy as monotherapy, HDACi show particular promise as combination partners with INT due to their ability to overcome tumor heterogeneity and immune evasion mechanisms[14-16]. The complementary actions of these two modalities—with cancer vaccines providing specific tumor antigens and HDACi enhancing their presentation and recognition—create a compelling rationale for combination therapy.
This review provides the first comprehensive analysis of the mechanistic rationale for integrating HDACi with INT as a next-generation combinatorial approach for cancer immunotherapy. We examine the synergistic effects, early clinical evidence, potential challenges, and future directions of this promising strategy that could trans-form treatment paradigms for patients with limited therapeutic options.”
Potential Coordination Mechanisms. The authors use a lot of specialized terms like "epigenetic reprogramming," "neoantigen presentation," "cGAS-STING pathway," and "class I HDAC inhibition" one after another. Short explanations will be provided after each technical term. Simple examples or small diagrams should be provided to make the complex ideas easier to understand.
Answer:
We are thankful to our reviewer for such thoughtful suggestion. We have added short explanations for those specialized terms, such as, for “class I HDAC inhibition”, we added explanation “(HDAC1, 2, 3, 8)” (Line 67 ).
Clinical Applications of HDACi Combined with Tumor Vaccines. This sections talks about early clinical trials combining HDAC inhibitors with vaccines (PVX-410, H1299 vaccine). However, the presence of limited clinical data makes it feel a bit speculative.
Answer:
Our sincere thanks are due to the reviewer for his/her meticulous and dedicated reviewing work. We have updated the section on the clinical application of HDACi combined with tumor vaccines. We have supplemented and improved the key details of the trial (such as patient numbers, endpoints, preliminary results)(Line 674-675). Table 2 has also been updated and has been marked in yellow. Table 2 and the revised clinical research content also has been shown as followed:
“4.1. Polypeptide Vaccine PVX-410 Combined with Citarinostat
Current clinical research on the combination of HDACi and tumor vaccines for cancer treatment remains limited, with most studies in early-phase clinical trials. For example, a Phase Ib clinical trial (NCT02886065) targeting smoldering multiple myeloma (SMM) is currently underway (Table 1). SMM is an asymptomatic clonal plasma cell proliferative disorder, intermediate between monoclonal gammopathy of undetermined significance (MGUS) and multiple myeloma (MM)[129].The study protocol includes two arms: the first arm combines a peptide-based vaccine (PVX-410) with Citarinostat (CC-96241), a small-molecule oral histone deacetylase inhibitor; the second arm employs a triple-drug regimen of PVX-410, Citarinostat, and lenalidomide (Table 2).
4.2. Multi-Drug Combination Regimens Beyond Dual Therapy
- Combination of H1299 Cell Lysate Vaccine, Entinostat (HDACi), Nivolumab, and Montanide(R) ISA-51 VG Adjuvant
A Phase I/II clinical trial (NCT05898828) aimed to evaluate the safety of a quad-ruple regimen comprising the H1299 cell lysate vaccine(Table 2), Entinostat (a histone deacetylase inhibitor, HDACi), Nivolumab (an anti-PD-1 antibody), and Montanide(R) ISA-51 VG adjuvant (an immunostimulatory adjuvant) in patients with advanced esophageal cancer (EsC). However, this study was withdrawn due to insufficient pa-tient enrollment (Table 2).
Previous clinical research (NCT02054104) demonstrated that the H1299 lysate vaccine reduced the percentage of regulatory T cells (Tregs) and downregulated PD-1 expression on Tregs (P=0.0027) in patients with primary thoracic malignancies, thereby enhancing anti-tumor immunity[132]. Another study revealed that Entinostat amplifies antigen-stimulated T-cell responses by suppressing immunosuppressive cell populations, including Tregs and monocytic myeloid-derived suppressor cells (M-MDSCs), further potentiating anti-tumor immunity[133].
- Combination of BN-Brachyury Vaccine, M7824, T-DM1, and Entinostat
A multi-drug regimen combining BN-Brachyury vaccine, M7824 (a novel bifunctional fusion protein), T-DM1 (an antibody-drug conjugate), and Entinostat (an HDAC inhibitor) has been investigated for metastatic breast cancer in a clinical trial (NCT04296942) (Table 2). The BN-Brachyury vaccine is a recombinant poxviral vaccine targeting the transcription factor brachyury, which is overexpressed in advanced cancers and associated with drug resistance, epithelial-mesenchymal transition (EMT), and metastatic potential. This trial was terminated due to emerging safety concerns related to M7824 and slow patient enrollment (Table 2). Since only one participant was in-cluded in this study, the overall response rate could not be calculated. The progres-sion/recurrent time of this included participant was approximately 5 months and 17 days.
A previous phase I study (NCT04134312) reported that BN-Brachyury vaccine administration in advanced solid tumor patients induced CD4+ and CD8+ T-cell re-sponses in 69% of patients. Notably, 88% and 64% of patients exhibited CD4+ and/or CD8+ T-cell reactivity against cascade antigens CEA and MUC1, respectively, which were not encoded by the vaccine. These T-cell responses were dose-dependent, suggesting the vaccine’s potential to activate anti-tumor immunity even in advanced disease[134].
In vitro studies demonstrated that Entinostat reduces phosphorylation of STAT3 and NFκB, thereby downregulating immunosuppressive downstream targets such as IL-6, IL-10, and Nox2. This mechanism enhances anti-tumor immune efficacy by alleviating immunosuppression[135].”
Table 2. Completed and ongoing clinical trials of HDACi in combination with tumor vaccines or other adjuvants.
|
Trial Number |
Launch |
Phase |
Study Status |
HDACi (targets) |
INT |
Other combined agents |
Cancer type |
Patient numbers |
Endpoints |
Preliminary results |
|
|
NCT02886065 |
2017 |
Ib |
Active |
Citarinostat
|
PVX-410 |
- |
Smoldering Multiple Myeloma |
19 |
Safety And Tolerability Of The Vaccine |
No Results Posted |
|
|
NCT05898828 |
2024 |
I/II |
Withdrawn |
Entinostat |
H1299 cell lysate vaccine |
Nivolumab, Montanide(R) ISA-51 VG |
Advanced esophageal Cancer |
0 |
Safe dose/frequency of immunologic responses |
No Results Posted |
|
|
NCT04296942 |
2021 |
I |
Terminated |
Entinostat |
BN-Brachyury vaccine |
M7824,T-DM1 |
Advanced Stage Breast Cancer |
1 |
Overall Response |
Progression/recurrentce time is 5 months and 17 days |
|
Current Issues and Challenges. This part discusses challenges like PD-L1 expression, safety issues, optimal drug sequencing, and need for biomarkers. Some sections are slightly repetitive (especially about PD-L1) that should be improved.
Answer:
We sincerely thank our reviewer for such careful and patient review. We fully agree with his/her comments and have rewritten and optimized Section 5.1. The new version effectively eliminates redundancies from the original text through a clear paragraph structure, concise language expression, and logically coherent content organization. Our team has endeavored through these revisions to more clearly present the complete logical chain from PD-L1's basic functions to its nuclear translocation mechanism, and further to the effects of HDAC inhibitors and their clinical applications, enabling readers to better understand the content of this section. The modified parts have been highlighted in yellow in the manuscript(Line 726-794) and are displayed below:
“5.1. Impact on PD-L1 Expression
While immunotherapy targeting programmed death-1 (PD-1) and its ligand PD-L1 has yielded remarkable clinical outcomes across various tumor types [136], only a subset of patients achieve durable responses. PD-L1 nuclear translocation has emerged as a critical factor limiting therapeutic efficacy [137]. Beyond its conventional immunosuppressive role at the plasma membrane, PD-L1 enhances tumor cell anti-apoptotic capabilities, promotes mTOR activity, and regulates glycolytic metabolism [138].
Research demonstrates that PD-L1 translocates to cell nuclei where it modulates inflammatory and immune responses, promotes tumor invasiveness and metastasis, and triggers expression of immune checkpoint molecules unaffected by PD-1/PD-L1 blockade, resulting in acquired resistance to immunotherapy [137, 139]. Gao's team conducted comprehensive studies using CD274 knock-off tumor cells for chromatin immunoprecipitation and sequencing (ChIP-seq), revealing that nuclear PD-L1 specifically triggers gene expression in immune response pathways. Nuclear PD-L1 positively correlates with immune response-related transcription factors including STAT3, RelA (p65), and c-Jun40, interacting with these factors on DNA to influence anti-tumor immunity[137].
Further molecular analyses revealed that reduced PD-L1 expression leads to downregulation of genes associated with immune surveillance evasion, such as PDCD1LG2 (encoding PD-L2), VSIR (encoding VISTA), and CD276 (encoding B7-H3). This confirms that nuclear PD-L1 upregulates multiple immune checkpoint genes in tumor cells, contributing to resistance against PD-L1/PD-1 blockade therapy[137]. Evidently, PD-L1 in the cell nucleus enhances the activation of multiple immune response pathways, thereby evading immune surveillance [137].
The nuclear translocation of PD-L1 is regulated by acetylation at the Lys 263 site in its C-tail, controlled by HDAC2 inhibitors [137]. Huntingtin-interacting protein 1-related protein (HIP1R) specifically interacts with the PD-L1 C-tail to initiate nuclear translocation. Both HIP1R expression and its binding capacity to PD-L1 depend on Lys 263 acetylation levels, which are regulated by HDAC2 expression [137, 140, 141].
Upon nuclear entry, PD-L1 interacts with DNA to regulate transcription of antigen presentation-related genes (including MHC-I-related genes) and inflammation path-way-related genes (including IFN-I-related genes). It simultaneously increases expression of other immune checkpoint genes like PD-L2 and VISTA, enhancing cytotoxic T lymphocyte exhaustion and impairing PD-L1 blockade efficacy[137]. More importantly, HDAC2 inhibitors not only inhibit nuclear PD-L1 translocation, enhancing the efficacy of PD-1/PD-L1 checkpoint inhibitors, but also significantly increase the proportion of CD8+ and CD8+GranB (granzyme B)+ cells within tumor-infiltrating lymphocytes. These inhibitors improve the CD8+ cytotoxic T cell to regulatory T cell (CD4+FOXP3+) ratio, modulate cytokine levels (including IL-4, IFN-γ, and TNF-α), and enhance the tumor immune microenvironment [142].
In uveal melanoma (UM) cells, HDAC2 inhibitors restore PD-L1 acetylation, pre-vent its nuclear entry, and inhibit p-STAT3 binding to the EGR1 promoter region, reducing EGR1 expression and suppressing angiogenic capacity. This suggests that combining anti-PD-L1 immunotherapy with HDAC2 inhibitors could attenuate tumor angiogenesis in UM, offering a novel therapeutic approach [143]. Studies in melanoma reveal that low expression of MIIP and high expression of its downstream molecule HDAC6 correlate with poorer overall survival, with a positive correlation between HDAC6 and PD-L1 protein expression. Experimental evidence indicates that the HDAC6/STAT3 axis regulates PD-L1 expression, suggesting that targeting the MIIP/HDAC6/PD-L1 pathway might enhance immune checkpoint inhibitor efficacy in melanoma [144, 145].
HDAC inhibitors that restore PD-L1 expression on tumor cells may potentially undermine mRNA cancer vaccine effectiveness [146]. Prolonged exposure to elevated PD-L1 levels might cause functional exhaustion of vaccine-induced T cells, diminishing their killing capacity [147]. Moreover, prolonged exposure to high levels of PD-L1 might lead to functional exhaustion of vaccine-induced T cells, reducing their killing capacity[148]. Additionally, PD-L1 may interfere with antigen-presenting cell function, affecting the effective presentation of mRNA vaccine-delivered antigens.
Despite these challenges, promising optimization strategies exist. Tumors with elevated PD-L1 expression may respond particularly well to combination therapy in-volving both mRNA cancer vaccines and PD-1/PD-L1 inhibitors[149]. Such combinations could overcome immunosuppression within the tumor microenvironment, enhancing therapeutic efficacy [149]. HDAC inhibitors could serve as "pretreatment" to ensure high PD-L1 expression on tumor cells, followed by sequential or concurrent administration of mRNA vaccines alongside PD-1/PD-L1 inhibitors. Through deeper mechanistic understanding, more effective triple combination immunotherapeutic approaches may improve clinical outcomes for cancer patients. Moreover, immune responses triggered by mRNA vaccines potentially increase IFN-γ levels within the tumor microenvironment, subsequently inducing elevated PD-L1 expression[150, 151]. This feedback mechanism suggests that dynamic changes in PD-L1 expression could serve as a clinically relevant predictive biomarker.”
Round 2
Reviewer 4 Report
Comments and Suggestions for Authors
The authors have responded to my comments in the revised manuscript.